# Statistical Optimality of Stochastic Gradient Descent on Hard Learning Problems through Multiple Passes

**Loucas Pillaud-Vivien**
INRIA - Ecole Normale Supérieure
PSL Research University
loucas.pillaud-vivien@inria.fr

**Alessandro Rudi**
INRIA - Ecole Normale Supérieure
PSL Research University
alessandro.rudi@inria.fr

**Francis Bach**
INRIA - Ecole Normale Supérieure
PSL Research University
francis.bach@inria.fr

## Abstract

We consider stochastic gradient descent (SGD) for least-squares regression with potentially several passes over the data. While several passes have been widely reported to perform practically better in terms of predictive performance on unseen data, the existing theoretical analysis of SGD suggests that a single pass is statistically optimal. While this is true for low-dimensional easy problems, we show that for hard problems, multiple passes lead to statistically optimal predictions while single pass does not; we also show that in these hard models, the optimal number of passes over the data increases with sample size. In order to define the notion of hardness and show that our predictive performances are optimal, we consider potentially infinite-dimensional models and notions typically associated to kernel methods, namely, the decay of eigenvalues of the covariance matrix of the features and the complexity of the optimal predictor as measured through the covariance matrix. We illustrate our results on synthetic experiments with non-linear kernel methods and on a classical benchmark with a linear model.

## 1 Introduction

Stochastic gradient descent (SGD) and its multiple variants—averaged [1], accelerated [2], variance-reduced [3, 4, 5]—are the workhorses of large-scale machine learning, because (a) these methods looks at only a few observations before updating the corresponding model, and (b) they are known in theory and in practice to generalize well to unseen data [6].

Beyond the choice of step-size (often referred to as the learning rate), the number of passes to make on the data remains an important practical and theoretical issue. In the context of finite-dimensional models (least-squares regression or logistic regression), the theoretical answer has been known for many years: a single passes suffices for the optimal statistical performance [1, 7]. Worse, most of the theoretical work only apply to single pass algorithms, with some exceptions leading to analyses of multiple passes when the step-size is taken smaller than the best known setting [8, 9].

However, in practice, multiple passes are always performed as they empirically lead to better generalization (e.g., loss on unseen test data) [6]. But no analysis so far has been able to show that, given the appropriate step-size, multiple pass SGD was theoretically better than single pass SGD.

The main contribution of this paper is to show that for least-squares regression, while single pass averaged SGD is optimal for a certain class of "easy" problems, multiple passes are needed to reach optimal prediction performance on another class of "hard" problems.

In order to define and characterize these classes of problems, we need to use tools from infinite-dimensional models which are common in the analysis of kernel methods. De facto, our analysis will be done in infinite-dimensional feature spaces, and for finite-dimensional problems where the dimension far exceeds the number of samples, using these tools are the only way to obtain non-vacuous dimension-independent bounds. Thus, overall, our analysis applies both to finite-dimensional models with explicit features (parametric estimation), and to kernel methods (non-parametric estimation).

The two important quantities in the analysis are:

(a) The decay of eigenvalues of the covariance matrix $\Sigma$ of the input features, so that the ordered eigenvalues $\lambda_m$ decay as $O(m^{-\alpha})$; the parameter $\alpha \geqslant 1$ characterizes the size of the feature space, $\alpha = 1$ corresponding to the largest feature spaces and $\alpha = +\infty$ to finite-dimensional spaces. The decay will be measured through $\mathrm{tr}\Sigma^{1/\alpha} = \sum_m \lambda_m^{1/\alpha}$, which is small when the decay of eigenvalues is faster than $O(m^{-\alpha})$.

(b) The complexity of the optimal predictor $\theta_*$ as measured through the covariance matrix $\Sigma$, that is with coefficients $\langle e_m, \theta_* \rangle$ in the eigenbasis $(e_m)_m$ of the covariance matrix that decay so that $\langle \theta_*, \Sigma^{1-2r}\theta_* \rangle$ is small. The parameter $r \geqslant 0$ characterizes the difficulty of the learning problem: $r = 1/2$ corresponds to characterizing the complexity of the predictor through the squared norm $\|\theta_*\|^2$, and thus $r$ close to zero corresponds to the hardest problems while $r$ larger, and in particular $r \geqslant 1/2$, corresponds to simpler problems.

Dealing with non-parametric estimation provides a simple way to evaluate the optimality of learning procedures. Indeed, given problems with parameters $r$ and $\alpha$, the best prediction performance (averaged square loss on unseen data) is well known [10] and decay as $O(n^{\frac{-2r\alpha}{2r\alpha+1}})$, with $\alpha = +\infty$ leading to the usual parametric rate $O(n^{-1})$. For *easy problems*, that is for which $r \geqslant \frac{\alpha-1}{2\alpha}$, then it is known that most iterative algorithms achieve this optimal rate of convergence (but with various running-time complexities), such as exact regularized risk minimization [11], gradient descent on the empirical risk [12], or averaged stochastic gradient descent [13].

We show that for *hard problems*, that is for which $r \leqslant \frac{\alpha-1}{2\alpha}$ (see Example 1 for a typical hard problem), then multiple passes are superior to a single pass. More precisely, under additional assumptions detailed in Section 2 that will lead to a subset of the hard problems, with $\Theta(n^{(\alpha-1-2r\alpha)/(1+2r\alpha)})$ passes, we achieve the optimal statistical performance $O(n^{\frac{-2r\alpha}{2r\alpha+1}})$, while for all other hard problems, a single pass only achieves $O(n^{-2r})$. This is illustrated in Figure 1.

We thus get a number of passes that grows with the number of observations $n$ and depends precisely on the quantities $r$ and $\alpha$. In synthetic experiments with kernel methods where $\alpha$ and $r$ are known, these scalings are precisely observed. In experiments on parametric models with large dimensions, we also exhibit an increasing number of required passes when the number of observations increases.

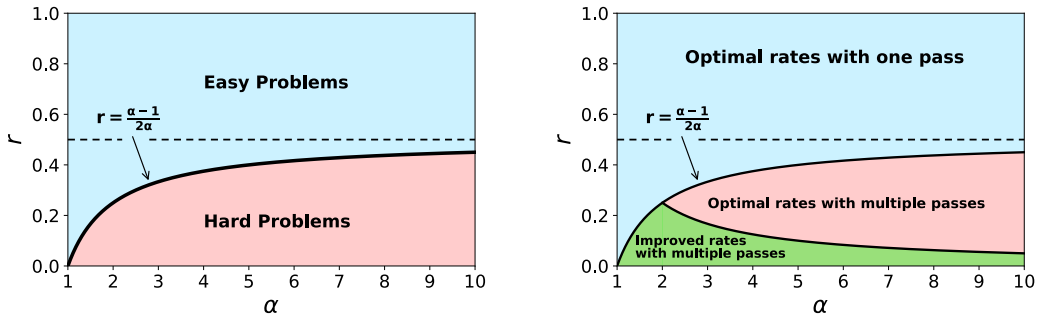

Figure 1 – (Left) easy and hard problems in the $(\alpha, r)$-plane. (Right) different regions for which multiple passes improved known previous bounds (green region) or reaches optimality (red region).

# 2 Least-squares regression in finite dimension

We consider a joint distribution $\rho$ on pairs of input/output $(x, y) \in \mathcal{X} \times \mathbb{R}$, where $\mathcal{X}$ is any input space, and we consider a feature map $\Phi$ from the input space $\mathcal{X}$ to a feature space $\mathcal{H}$, which we assume Euclidean in this section, so that all quantities are well-defined. In Section 4, we will extend all the notions to Hilbert spaces.

## 2.1 Main assumptions

We are considering predicting $y$ as a linear function $f_\theta(x) = \langle \theta, \Phi(x) \rangle_{\mathcal{H}}$ of $\Phi(x)$, that is estimating $\theta \in \mathcal{H}$ such that $F(\theta) = \frac{1}{2}\mathbb{E}(y - \langle \theta, \Phi(x) \rangle_{\mathcal{H}})^2$ is as small as possible. Estimators will depend on $n$ observations, with standard sampling assumptions:

**(A1)**    *The $n$ observations $(x_i, y_i) \in \mathcal{X} \times \mathbb{R}$, $i = 1, \ldots, n$, are independent and identically distributed from the distribution $\rho$.*

Since $\mathcal{H}$ is finite-dimensional, $F$ always has a (potentially non-unique) minimizer in $\mathcal{H}$ which we denote $\theta_*$. We make the following standard boundedness assumptions:

**(A2)**    $\|\Phi(x)\| \leqslant R$ *almost surely, $|y - \langle \theta_*, \Phi(x) \rangle_{\mathcal{H}}|$ is almost surely bounded by $\sigma$ and $|y|$ is almost surely bounded by $M$.*

In order to obtain improved rates with multiple passes, and motivated by the equivalent previously used condition in reproducing kernel Hilbert spaces presented in Section 4, we make the following extra assumption (we denote by $\Sigma = \mathbb{E}[\Phi(x) \otimes_{\mathcal{H}} \Phi(x)]$ the (non-centered) covariance matrix).

**(A3)**    *For $\mu \in [0, 1]$, there exists $\kappa_\mu \geqslant 0$ such that, almost surely, $\Phi(x) \otimes_{\mathcal{H}} \Phi(x) \preccurlyeq_{\mathcal{H}} \kappa_\mu^2 R^{2\mu} \Sigma^{1-\mu}$. Note that it can also be written as $\|\Sigma^{\mu/2 - 1/2} \Phi(x)\|_{\mathcal{H}} \leqslant \kappa_\mu R^\mu$.*

Assumption (A3) is always satisfied with any $\mu \in [0, 1]$, and has particular values for $\mu = 1$, with $\kappa_1 = 1$, and $\mu = 0$, where $\kappa_0$ has to be larger than the dimension of the space $\mathcal{H}$.

We will also introduce a parameter $\alpha$ that characterizes the decay of eigenvalues of $\Sigma$ through the quantity $\mathrm{tr}\Sigma^{1/\alpha}$, as well as the difficulty of the learning problem through $\|\Sigma^{1/2-r}\theta_*\|_{\mathcal{H}}$, for $r \in [0, 1]$. In the finite-dimensional case, these quantities can always be defined and most often finite, but may be very large compared to sample size. In the following assumptions the quantities are assumed to be finite and small compared to $n$.

**(A4)**    *There exists $\alpha > 1$ such that $\mathrm{tr}\, \Sigma^{1/\alpha} < \infty$.*

Assumption (A4) is often called the "capacity condition". First note that this assumption implies that the decreasing sequence of the eigenvalues of $\Sigma$, $(\lambda_m)_{m \geqslant 1}$, satisfies $\lambda_m = o\left(1/m^\alpha\right)$. Note that $\mathrm{tr}\Sigma^\mu \leqslant \kappa_\mu^2 R^{2\mu}$ and thus often we have $\mu \geqslant 1/\alpha$, and in the most favorable cases in Section 4, this bound will be achieved. We also assume:

**(A5)**    *There exists $r \geqslant 0$, such that $\|\Sigma^{1/2-r}\theta_*\|_{\mathcal{H}} < \infty$.*

Assumption (A5) is often called the "source condition". Note also that for $r = 1/2$, this simply says that the optimal predictor has a small norm.

In the subsequent sections, we essentially assume that $\alpha$, $\mu$ and $r$ are chosen (by the theoretical analysis, not by the algorithm) so that all quantities $R_\mu$, $\|\Sigma^{1/2-r}\theta_*\|_{\mathcal{H}}$ and $\mathrm{tr}\Sigma^{1/\alpha}$ are finite and small. As recalled in the introduction, these parameters are often used in the non-parametric literature to quantify the hardness of the learning problem (Figure 1).

We will use result with $O(\cdot)$ and $\Theta(\cdot)$ notations, which will all be independent of $n$ and $t$ (number of observations and number of iterations) but can depend on other finite constants. Explicit dependence on all parameters of the problem is given in proofs. More precisely, we will use the usual $O(\cdot)$ and $\Theta(\cdot)$ notations for sequences $b_{nt}$ and $a_{nt}$ that can depend on $n$ and $t$, as $a_{nt} = O(b_{nt})$ if and only if, there exists $M > 0$ such that for all $n, t$, $a_{nt} \leqslant Mb_{nt}$, and $a_{nt} = \Theta(b_{nt})$ if and only if, there exist $M, M' > 0$ such that for all $n, t$, $M'b_{nt} \leqslant a_{nt} \leqslant Mb_{nt}$.

## 2.2 Related work

Given our assumptions above, several algorithms have been developed for obtaining low values of the expected excess risk $\mathbb{E}\big[F(\theta)\big] - F(\theta_*)$.

**Regularized empirical risk minimization.** Forming the empirical risk $\hat{F}(\theta)$, it minimizes $\hat{F}(\theta) + \lambda\|\theta\|_{\mathcal{H}}^2$, for appropriate values of $\lambda$. It is known that for easy problems where $r \geqslant \frac{\alpha-1}{2\alpha}$, it achieves the optimal rate of convergence $O(n^{\frac{-2r\alpha}{2r\alpha+1}})$ [11]. However, algorithmically, this requires to solve a linear system of size $n$ times the dimension of $\mathcal{H}$. One could also use fast variance-reduced stochastic gradient algorithms such as SAG [3], SVRG [4] or SAGA [5], with a complexity proportional to the dimension of $\mathcal{H}$ times $n + R^2/\lambda$.

**Early-stopped gradient descent on the empirical risk.** Instead of solving the linear system directly, one can use gradient descent with early stopping [12, 14]. Similarly to the regularized empirical risk minimization case, a rate of $O(n^{-\frac{2r\alpha}{2r\alpha+1}})$ is achieved for the easy problems, where $r \geqslant \frac{\alpha-1}{2\alpha}$. Different iterative regularization techniques beyond batch gradient descent with early stopping have been considered, with computational complexities ranging from $O(n^{1+\frac{\alpha}{2r\alpha+1}})$ to $O(n^{1+\frac{\alpha}{4r\alpha+2}})$ times the dimension of $\mathcal{H}$ (or $n$ in the kernel case in Section 4) for optimal predictions [12, 15, 16, 17, 14].

**Stochastic gradient.** The usual stochastic gradient recursion is iterating from $i = 1$ to $n$,

$$\theta_i = \theta_{i-1} + \gamma\big(y_i - \langle\theta_{i-1}, \Phi(x_i)\rangle_{\mathcal{H}}\big)\Phi(x_i),$$

with the averaged iterate $\bar{\theta}_n = \frac{1}{n}\sum_{i=1}^n \theta_i$. Starting from $\theta_0 = 0$, [18] shows that the expected excess performance $\mathbb{E}[F(\bar{\theta}_n)] - F(\theta_*)$ decomposes into a *variance* term that depends on the noise $\sigma^2$ in the prediction problem, and a *bias term*, that depends on the deviation $\theta_* - \theta_0 = \theta_*$ between the initialization and the optimal predictor. Their bound is, up to universal constants, $\frac{\sigma^2\dim(\mathcal{H})}{n} + \frac{\|\theta_*\|_{\mathcal{H}}^2}{\gamma n}$.

Further, [13] considered the quantities $\alpha$ and $r$ above to get the bound, up to constant factors:

$$\frac{\sigma^2\mathrm{tr}\Sigma^{1/\alpha}(\gamma n)^{1/\alpha}}{n} + \frac{\|\Sigma^{1/2-r}\theta_*\|^2}{\gamma^{2r}n^{2r}}.$$

We recover the finite-dimensional bound for $\alpha = +\infty$ and $r = 1/2$. The bounds above are valid for all $\alpha \geqslant 1$ and all $r \in [0, 1]$, and the step-size $\gamma$ is such that $\gamma R^2 \leqslant 1/4$, and thus we see a natural trade-off appearing for the step-size $\gamma$, between bias and variance.

When $r \geqslant \frac{\alpha-1}{2\alpha}$, then the optimal step-size minimizing the bound above is $\gamma \propto n^{\frac{-2\alpha\min\{r,1\}-1+\alpha}{2\alpha\min\{r,1\}+1}}$, and the obtained rate is optimal. Thus a single pass is optimal. However, when $r \leqslant \frac{\alpha-1}{2\alpha}$, the best step-size does not depend on $n$, and one can only achieve $O(n^{-2r})$.

Finally, in the same multiple pass set-up as ours, [9] has shown that for easy problems where $r \geqslant \frac{\alpha-1}{2\alpha}$ (and single-pass averaged SGD is already optimal) that multiple-pass non-averaged SGD is becoming optimal after a correct number of passes (while single-pass is not). Our proof principle of comparing to batch gradient is taken from [9], but we apply it to harder problems where $r \leqslant \frac{\alpha-1}{2\alpha}$. Moreover we consider the multi-pass averaged-SGD algorithm, instead of non-averaged SGD, and take explicitly into account the effect of Assumption (A3).

## 3 Averaged SGD with multiple passes

We consider the following algorithm, which is stochastic gradient descent with sampling with replacement with multiple passes over the data (we experiment in Section E of the Appendix with cycling over the data, with or without reshuffling between each pass).

- **Initialization**: $\theta_0 = \bar{\theta}_0 = 0$, $t$ = maximal number of iterations, $\gamma = 1/(4R^2)$ = step-size
- **Iteration**: for $u = 1$ to $t$, sample $i(u)$ uniformly from $\{1, \ldots, n\}$ and make the step

$$\theta_u = \theta_{u-1} + \gamma\big(y_{i(u)} - \langle\theta_{t-1}, \Phi(x_{i(u)})\rangle_{\mathcal{H}}\big)\Phi(x_{i(u)}) \quad \text{and} \quad \bar{\theta}_u = (1 - \tfrac{1}{u})\bar{\theta}_{u-1} + \tfrac{1}{u}\theta_u.$$

In this paper, following [18, 13], but as opposed to [19], we consider unregularized recursions. This removes a unnecessary regularization parameter (at the expense of harder proofs).

## 3.1 Convergence rate and optimal number of passes

Our main result is the following (see full proof in Appendix):

**Theorem 1.** *Let $n \in \mathbb{N}^*$ and $t \geqslant n$, under Assumptions (A1), (A2), (A3), (A4), (A5), (A6), with $\gamma = 1/(4R^2)$.*

- *For $\mu\alpha < 2r\alpha + 1 < \alpha$, if we take $t = \Theta(n^{\alpha/(2r\alpha+1)})$, we obtain the following rate:*

$$\mathbb{E}F(\bar{\theta}_t) - F(\theta_*) = O(n^{-2r\alpha/(2r\alpha+1)}).$$

- *For $\mu\alpha \geqslant 2r\alpha + 1$, if we take $t = \Theta(n^{1/\mu} (\log n)^{\frac{1}{\mu}})$, we obtain the following rate:*

$$\mathbb{E}F(\bar{\theta}_t) - F(\theta_*) \leqslant O(n^{-2r/\mu}).$$

**Sketch of proof.** The main difficulty in extending proofs from the single pass case [18, 13] is that as soon as an observation is processed twice, then statistical dependences are introduced and the proof does not go through. In a similar context, some authors have considered stability results [8], but the large step-sizes that we consider do not allow this technique. Rather, we follow [16, 9] and compare our multi-pass stochastic recursion $\theta_t$ to the batch gradient descent iterate $\eta_t$ defined as $\eta_t = \eta_{t-1} + \frac{\gamma}{n}\sum_{i=1}^{n}\left(y_i - \langle\eta_{t-1}, \Phi(x_i)\rangle_{\mathcal{H}}\right)\Phi(x_i)$ with its averaged iterate $\bar{\eta}_t$. We thus need to study the predictive performance of $\bar{\eta}_t$ and the deviation $\bar{\theta}_t - \bar{\eta}_t$. It turns out that, given the data, the deviation $\theta_t - \eta_t$ satisfies an SGD recursion (with the respect to the randomness of the sampling with replacement). For a more detailed summary of the proof technique see Section B.

The novelty compared to [16, 9] is (a) to use refined results on averaged SGD for least-squares, in particular convergence in various norms for the deviation $\bar{\theta}_t - \bar{\eta}_t$ (see Section A), that can use our new Assumption (A3). Moreover, (b) we need to extend the convergence results for the batch gradient descent recursion from [14], also to take into account the new assumption (see Section D). These two results are interesting on their own.

**Improved rates with multiple passes.** We can draw the following conclusions:

- If $2\alpha r + 1 \geqslant \alpha$, that is, easy problems, it has been shown by [13] that a single pass with a smaller step-size than the one we propose here is optimal, and our result does not apply.

- If $\mu\alpha < 2r\alpha + 1 < \alpha$, then our proposed number of iterations is $t = \Theta(n^{\alpha/(2\alpha r+1)})$, which is now greater than $n$; the convergence rate is then $O(n^{\frac{-2r\alpha}{2r\alpha+1}})$, and, as we will see in Section 4.2, the predictive performance is then optimal when $\mu \leqslant 2r$.

- If $\mu\alpha \geqslant 2r\alpha + 1$, then with a number of iterations is $t = \Theta(n^{1/\mu})$, which is greater than $n$ (thus several passes), with a convergence rate equal to $O(n^{-2r/\mu})$, which improves upon the best known rates of $O(n^{-2r})$. As we will see in Section 4.2, this is not optimal.

Note that these rates are theoretically only bounds on the optimal number of passes over the data, and one should be cautious when drawing conclusions; however our simulations on synthetic data, see Figure 2 in Section 5, confirm that our proposed scalings for the number of passes is observed in practice.

## 4 Application to kernel methods

In the section above, we have assumed that $\mathcal{H}$ was finite-dimensional, so that the optimal predictor $\theta_* \in \mathcal{H}$ was always defined. Note however, that our bounds that depends on $\alpha$, $r$ and $\mu$ are *independent of the dimension*, and hence, intuitively, following [19], should apply immediately to infinite-dimensional spaces.

We now first show in Section 4.1 how this intuition can be formalized and how using kernel methods provides a particularly interesting example. Moreover, this interpretation allows to characterize the statistical optimality of our results in Section 4.2.

## 4.1 Extension to Hilbert spaces, kernel methods and non-parametric estimation

Our main result in Theorem 1 extends directly to the case where $\mathcal{H}$ is an infinite-dimensional Hilbert space. In particular, given a feature map $\Phi : \mathcal{X} \to \mathcal{H}$, any vector $\theta \in \mathcal{H}$ is naturally associated to a function defined as $f_\theta(x) = \langle \theta, \Phi(x) \rangle_{\mathcal{H}}$. Algorithms can then be run with infinite-dimensional objects if the *kernel* $K(x', x) = \langle \Phi(x'), \Phi(x) \rangle_{\mathcal{H}}$ can be computed efficiently. This identification of elements $\theta$ of $\mathcal{H}$ with functions $f_\theta$ endows the various quantities we have introduced in the previous sections, with natural interpretations in terms of functions. The stochastic gradient descent described in Section 3 adapts instantly to this new framework as the iterates $(\theta_u)_{u \leqslant t}$ are linear combinations of feature vectors $\Phi(x_i)$, $i = 1, \ldots, n$, and the algorithms can classically be "kernelized" [20, 13], with an overall running time complexity of $O(nt)$.

First note that Assumption (A3) is equivalent to, for all $x \in \mathcal{X}$ and $\theta \in \mathcal{H}$, $|f_\theta(x)|^2 \leqslant \kappa_\mu^2 R^{2\mu} \langle f_\theta, \Sigma^{1-\mu} f_\theta \rangle_{\mathcal{H}}$, that is, $\|g\|_{L_\infty}^2 \leqslant \kappa_\mu^2 R^{2\mu} \|\Sigma^{1/2-\mu/2} g\|_{\mathcal{H}}^2$ for any $g \in \mathcal{H}$ and also implies[1] $\|g\|_{L_\infty} \leqslant \kappa_\mu R^\mu \|g\|_{\mathcal{H}}^\mu \|g\|_{L_2}^{1-\mu}$, which are common assumptions in the context of kernel methods [22], essentially controlling in a more refined way the regularity of the whole space of functions associated to $\mathcal{H}$, with respect to the $L^\infty$-norm, compared to the too crude inequality $\|g\|_{L^\infty} = \sup_x |\langle \Phi(x), g \rangle_{\mathcal{H}}| \leqslant \sup_x \|\Phi(x)\|_{\mathcal{H}} \|g\|_{\mathcal{H}} \leqslant R \|g\|_{\mathcal{H}}$.

The natural relation with functions allows to analyze effects that are crucial in the context of learning, but difficult to grasp in the finite-dimensional setting. Consider the following prototypical example of a hard learning problem,

**Example 1** (Prototypical hard problem on simple Sobolev space). *Let $\mathcal{X} = [0,1]$, with $x$ sampled uniformly on $X$ and*

$$y = sign(x - 1/2) + \epsilon, \quad \Phi(x) = \{|k|^{-1} e^{2ik\pi x}\}_{k \in \mathbb{Z}^*}.$$

This corresponds to the kernel $K(x, y) = \sum_{k \in \mathbb{Z}^*} |k|^{-2} e^{2ik\pi(x-y)}$, which is well defined (and lead to the simplest Sobolev space). Note that for any $\theta \in \mathcal{H}$, which is here identified as the space of square-summable sequences $\ell^2(\mathbb{Z})$, we have $f_\theta(x) = \langle \theta, \Phi(x) \rangle_{\ell^2(\mathbb{Z})} = \sum_{k \in \mathbb{Z}^*} \frac{\theta_k}{|k|} e^{2ik\pi x}$. This means that for any estimator $\hat\theta$ given by the algorithm, $f_{\hat\theta}$ is at least once continuously differentiable, while the target function $sign(\cdot - 1/2)$ is not even continuous. Hence, we are in a situation where $\theta_*$, the minimizer of the excess risk, does not belong to $\mathcal{H}$. Indeed let represent $sign(\cdot - 1/2)$ in $\mathcal{H}$, for almost all $x \in [0,1]$, by its Fourier series $sign(x - 1/2) = \sum_{k \in \mathbb{Z}_*} \alpha_k e^{2ik\pi x}$, with $|\alpha_k| \sim 1/k$, an informal reasoning would lead to $(\theta_*)_k = \alpha_k |k| \sim 1$, which is not square-summable and thus $\theta_* \notin \mathcal{H}$. For more details, see [23, 24].

This setting generalizes important properties that are valid for Sobolev spaces, as shown in the following example, where $\alpha, r, \mu$ are characterized in terms of the smoothness of the functions in $\mathcal{H}$, the smoothness of $f^*$ and the dimensionality of the input space $\mathcal{X}$.

**Example 2** (Sobolev Spaces [25, 22, 26, 10]). *Let $\mathcal{X} \subseteq \mathbb{R}^d$, $d \in \mathbb{N}$, with $\rho_{\mathcal{X}}$ supported on $\mathcal{X}$, absolutely continous with the uniform distribution and such that $\rho_{\mathcal{X}}(x) \geqslant a > 0$ almost everywhere, for a given $a$. Assume that $f^*(x) = \mathbb{E}[y|x]$ is $s$-times differentiable, with $s > 0$. Choose a kernel, inducing Sobolev spaces of smoothness $m$ with $m > d/2$, as the Matérn kernel*

$$K(x', x) = \|x' - x\|^{m-d/2} \mathcal{K}_{d/2-m}(\|x' - x\|),$$

*where $\mathcal{K}_{d/2-m}$ is the modified Bessel function of the second kind. Then the assumptions are satisfied for any $\epsilon > 0$, with $\alpha = \frac{2m}{d}$, $\mu = \frac{d}{2m} + \epsilon$, $r = \frac{s}{2m}$.*

In the following subsection we compare the rates obtained in Thm. 1, with known lower bounds under the same assumptions.

## 4.2 Minimax lower bounds

In this section we recall known lower bounds on the rates for classes of learning problems satisfying the conditions in Sect. 2.1. Interestingly, the comparison below shows that our results in Theorem 1

are optimal in the setting $2r \geqslant \mu$. While the optimality of SGD was known for the regime $\{2r\alpha+1 \geqslant \alpha \cap 2r \geqslant \mu\}$, here we extend the optimality to the new regime $\alpha \geqslant 2r\alpha + 1 \geqslant \mu\alpha$, covering essentially all the region $2r \geqslant \mu$, as it is possible to observe in Figure 1, where for clarity we plotted the best possible value for $\mu$ that is $\mu = 1/\alpha$ [10] (which is true for Sobolev spaces).

When $r \in (0,1]$ is fixed, but there are no assumptions on $\alpha$ or $\mu$, then the optimal minimax rate of convergence is $O(n^{-2r/(2r+1)})$, attained by regularized empirical risk minimization [11] and other spectral filters on the empirical covariance operator [27].

When $r \in (0,1]$ and $\alpha \geqslant 1$ are fixed (but there are no constraints on $\mu$), the optimal minimax rate of convergence $O(n^{\frac{-2r\alpha}{2r\alpha+1}})$ is attained when $r \geqslant \frac{\alpha-1}{2\alpha}$, with empirical risk minimization [14] or stochastic gradient descent [13].

When $r \geqslant \frac{\alpha-1}{2\alpha}$, the rate of convergence $O(n^{\frac{-2r\alpha}{2r\alpha+1}})$ is known to be a lower bound on the optimal minimax rate, but the best upper-bound so far is $O(n^{-2r})$ and is achieved by empirical risk minimization [14] or stochastic gradient descent [13], and the optimal rate is not known.

When $r \in (0,1]$, $\alpha \geqslant 1$ and $\mu \in [1/\alpha, 1]$ are fixed, then the rate of convergence $O(n^{\frac{-\max\{\mu, 2r\}\alpha}{2\max\{\mu, 2r\}\alpha+1}})$ is known to be a lower bound on the optimal minimax rate [10]. This is attained by regularized empirical risk minimization when $2r \geqslant \mu$ [10], and *now by SGD with multiple passes*, and it is thus the optimal rate in this situation. When $2r < \mu$, the only known upper bound is $O(n^{-2\alpha r/(\mu\alpha+1)})$, and the optimal rate is not known.

## 5    Experiments

In our experiments, the main goal is to show that with more that one pass over the data, we can improve the accuracy of SGD when the problem is hard. We also want to highlight our dependence of the optimal number of passes (that is $t/n$) with respect to the number of observations $n$.

**Synthetic experiments.**    Our main experiments are performed on artificial data following the setting in [21]. For this purpose, we take kernels $K$ corresponding to splines of order $q$ (see [24]) that fulfill Assumptions (A1) (A2) (A3) (A4) (A5) (A6). Indeed, let us consider the following function

$$\Lambda_q(x,z) = \sum_{k \in \mathbb{Z}} \frac{e^{2i\pi k(x-z)}}{|k|^q},$$

defined almost everywhere on $[0,1]$, with $q \in \mathbb{R}$, and for which we have the interesting relationship: $\langle \Lambda_q(x, \cdot), \Lambda_{q'}(z, \cdot) \rangle_{L_2(d\rho_\mathcal{X})} = \Lambda_{q+q'}(x, z)$ for any $q, q' \in \mathbb{R}$. Our setting is the following:

- **Input distribution:** $\mathcal{X} = [0,1]$ and $\rho_\mathcal{X}$ is the uniform distribution.
- **Kernel:** $\forall (x,z) \in [0,1]$, $K(x,z) = \Lambda_\alpha(x,z)$.
- **Target function:** $\forall x \in [0,1]$, $\theta_* = \Lambda_{r\alpha+\frac{1}{2}}(x, 0)$.
- **Output distribution :** $\rho(y|x)$ is a Gaussian with variance $\sigma^2$ and mean $\theta_*$.

For this setting we can show that the learning problem satisfies Assumptions (A1) (A2) (A3) (A4) (A5) (A6) with $r$, $\alpha$, and $\mu = 1/\alpha$. We take different values of these parameters to encounter all the different regimes of the problems shown in Figure 1.

For each $n$ from 100 to 1000, we found the optimal number of steps $t_*(n)$ that minimizes the test error $F(\bar{\theta}_t) - F(\theta_*)$. Note that because of overfitting the test error increases for $t > t_*(n)$. In Figure 2, we show $t_*(n)$ with respect to $n$ in $\log$ scale. As expected, for the easy problems (where $r \geqslant \frac{\alpha-1}{2\alpha}$, see top left and right plots), the slope of the plot is 1 as one pass over the data is enough: $t_*(n) = \Theta(n)$. But we see that for hard problems (where $r \leqslant \frac{\alpha-1}{2\alpha}$, see bottom left and right plots), we need more than one pass to achieve optimality as the optimal number of iterations is very close to $t_*(n) = \Theta\left(n^{\frac{\alpha}{2r\alpha+1}}\right)$. That matches the theoretical predictions of Theorem 1. We also notice in the plots that, the bigger $\frac{\alpha}{2r\alpha+1}$ the harder the problem is and the bigger the number of epochs we have to take. Note, that to reduce the noise on the estimation of $t_*(n)$, plots show an average over 100 replications.

To conclude, the experiments presented in the section correspond exactly to the theoretical setting of the article (sampling with replacement), however we present in Figures 4 and 5 of Section E of the Appendix results on the same datasets for two different ways of sampling the data: (a)*without replacement*: for which we select randomly the data points but never use twice the same point in one epoch, (b) *cycles*: for which we pick successively the data points in the same order. The obtained scalings relating number of iterations or passes to number of observations are the same.

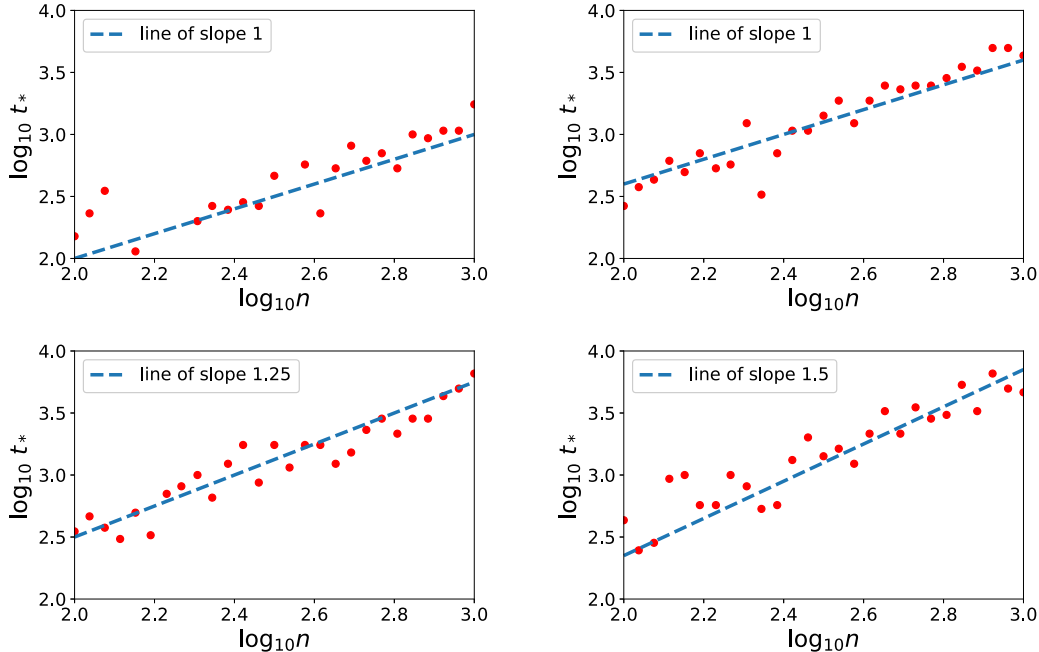

Figure 2 – The four plots represent each a different configuration on the $(\alpha, r)$ plan represented in Figure 1, for $r = 1/(2\alpha)$. **Top left** ($\alpha = 1.5$) and **right** ($\alpha = 2$) are two easy problems (Top right is the limiting case where $r = \frac{\alpha-1}{2\alpha}$) for which one pass over the data is optimal. **Bottom left** ($\alpha = 2.5$) and **right** ($\alpha = 3$) are two hard problems for which an increasing number of passes is required. The blue dotted line are the slopes predicted by the theoretical result in Theorem 1.

**Linear model.** To illustrate our result with some real data, we show how the optimal number of passes over the data increases with the number of samples. In Figure 3, we simply performed linear least-squares regression on the MNIST dataset and plotted the optimal number of passes over the data that leads to the smallest error on the test set. Evaluating $\alpha$ and $r$ from Assumptions (A4) and (A5), we found $\alpha = 1.7$ and $r = 0.18$. As $r = 0.18 \leqslant \frac{\alpha-1}{2\alpha} \sim 0.2$, Theorem 1 indicates that this corresponds to a situation where only one pass on the data is not enough, confirming the behavior of Figure 3. This suggests that learning MNIST with linear regression is a *hard problem*.

## 6 Conclusion

In this paper, we have shown that for least-squares regression, in hard problems where single-pass SGD is not statistically optimal ($r < \frac{\alpha-1}{2\alpha}$), then multiple passes lead to statistical optimality with a number of passes that somewhat surprisingly needs to grow with sample size, with a convergence rate which is superior to previous analyses of stochastic gradient. Using a non-parametric estimation, we show that under certain conditions ($2r \geqslant \mu$), we attain statistical optimality.

Our work could be extended in several ways: (a) our experiments suggest that cycling over the data and cycling with random reshuffling perform similarly to sampling with replacement, it would be interesting to combine our theoretical analysis with work aiming at analyzing other sampling schemes [28, 29]. (b) Mini-batches could be also considered with a potentially interesting effects compared to the streaming setting. Also, (c) our analysis focuses on least-squares regression, an extension to all smooth loss functions would widen its applicability. Moreover, (d) providing optimal

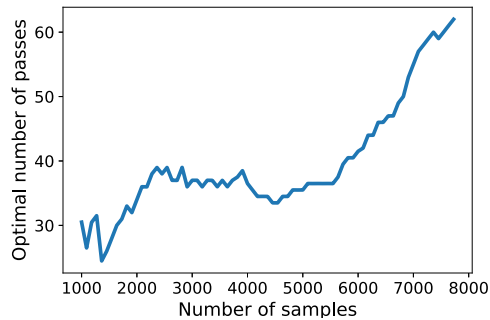

Figure 3 – For the MNIST data set, we show the optimal number of passes over the data with respect to the number of samples in the case of the linear regression.

efficient algorithms for the situation $2r < \mu$ is a clear open problem (for which the optimal rate is not known, even for non-efficient algorithms). Additionally, (e) in the context of classification, we could combine our analysis with [30] to study the potential discrepancies between training and testing losses and errors when considering high-dimensional models [31]. More generally, (f) we could explore the effect of our analysis for methods based on the least squares estimator in the context of structured prediction [32, 33, 34] and (non-linear) multitask learning [35]. Finally, (g) to reduce the computational complexity of the algorithm, while retaining the (optimal) statistical guarantees, we could combine multi-pass stochastic gradient descent, with approximation techniques like *random features* [36], extending the analysis of [37] to the more general setting considered in this paper.

**Acknowledgements**

We acknowledge support from the European Research Council (grant SEQUOIA 724063). We also thank Raphaël Berthier and Yann Labbé for their enlightening advices on this project.

## Footnotes

[1]Indeed, for any $g \in \mathcal{H}$, $\|\Sigma^{1/2-\mu/2} g\|_{\mathcal{H}} = \|\Sigma^{-\mu/2} g\|_{L_2} \leqslant \|\Sigma^{-1/2} g\|_{L_2}^\mu \|g\|_{L_2}^{1-\mu} = \|g\|_{\mathcal{H}}^\mu \|g\|_{L_2}^{1-\mu}$, where we used that for any $g \in \mathcal{H}$, any bounded operator $A$, $s \in [0,1]$: $\|A^s g\|_{L_2} \leqslant \|Ag\|_{L_2}^s \|g\|_{L_2}^{1-s}$ (see [21]).

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
