[Supplementary Material]

# Appendix

The appendix in constructed as follows:

- We first present in Section A a new result for stochastic gradient recursions which generalizes the work of [18] and [13] to more general norms. This result could be used in other contexts.

- The proof technique for Theorem 1 is presented in Section B.

- In Section C we give a proof of the various lemmas needed in the first part of the proof of Theorem 1 (deviation between SGD and batch gradient descent).

- In Section D we provide new results for the analysis of batch gradient descent, which are adapted to our new (A3), and instrumental in proving Theorem 1 in Section B.

- Finally, in Section E we present experiments for different sampling techniques.

## A    A general result for the SGD variance term

Independently of the problem studied in this paper, we consider i.i.d. observations $(z_t, \xi_t) \in \mathcal{H} \times \mathcal{H}$ a Hilbert space, and the recursion started from $\mu_0 = 0$.

$$\mu_t = (I - \gamma z_t \otimes z_t)\, \mu_{t-1} + \gamma \xi_t \tag{1}$$

(this will applied with $z_t = \Phi(x_{i(t)})$). This corresponds to the variance term of SGD. We denote by $\bar{\mu}_t$ the averaged iterate $\bar{\mu}_t = \frac{1}{t} \sum_{i=1}^{t} \mu_i$.

The goal of the proposition below is to provide a bound on $\mathbb{E}\left[\left\|H^{u/2}\bar{\mu}_t\right\|^2\right]$ for $u \in [0, \frac{1}{\alpha} + 1]$, where $H = \mathbb{E}\left[z_t \otimes z_t\right]$ is such that $\mathrm{tr}H^{1/\alpha}$ is finite. Existing results only cover the case $u = 1$.

**Proposition 1** (A general result for the SGD variance term). *Let us consider the recursion in Eq. (1) started at $\mu_0 = 0$. Denote $\mathbb{E}\left[z_t \otimes z_t\right] = H$, assume that $\mathrm{tr}H^{1/\alpha}$ is finite, $\mathbb{E}\left[\xi_t\right] = 0$, $\mathbb{E}\left[(z_t \otimes z_t)^2\right] \preccurlyeq R^2 H$, $\mathbb{E}\left[\xi_t \otimes \xi_t\right] \preccurlyeq \sigma^2 H$ and $\gamma R^2 \leqslant 1/4$, then for $u \in [0, \frac{1}{\alpha} + 1]$:*

$$\mathbb{E}\left[\left\|H^{u/2}\bar{\mu}_t\right\|^2\right] \leqslant 4\sigma^2 \gamma^{1-u}\, \frac{\gamma^{1/\alpha}\mathrm{tr}H^{1/\alpha}}{t^{u-1/\alpha}}. \tag{2}$$

### A.1    Proof principle

We follow closely the proof technique of [18], and prove Proposition 1 by showing it first for a "semi-stochastic" recursion, where $z_t \otimes z_t$ is replaced by its expectation (see Lemma 1). We will then compare our general recursion to the semi-stochastic one.

### A.2    Semi-stochastic recursion

**Lemma 1** (Semi-stochastic SGD). *Let us consider the following recursion $\mu_t = (I - \gamma H)\, \mu_{t-1} + \gamma \xi_t$ started at $\mu_0 = 0$. Assume that $\mathrm{tr}H^{1/\alpha}$ is finite, $\mathbb{E}\left[\xi_t\right] = 0$, $\mathbb{E}\left[\xi_t \otimes \xi_t\right] \preccurlyeq \sigma^2 H$ and $\gamma H \preccurlyeq I$, then for $u \in [0, \frac{1}{\alpha} + 1]$:*

$$\mathbb{E}\left[\left\|H^{u/2}\bar{\mu}_t\right\|^2\right] \leqslant \sigma^2 \gamma^{1-u}\, \gamma^{1/\alpha}\mathrm{tr}H^{1/\alpha}t^{1/\alpha-u}. \tag{3}$$

*Proof.* For $t \geqslant 1$ and $u \in [0, \frac{1}{\alpha} + 1]$, using an explicit formula for $\mu_t$ and $\bar{\mu}_t$ (see [18] for details), we get:

$$\mu_t = (I - \gamma H) \mu_{t-1} + \gamma \xi_t = (I - \gamma H)^t \mu_0 + \gamma \sum_{k=1}^{t} (I - \gamma H)^{t-k} \xi_k$$

$$\bar{\mu}_t = \frac{1}{t} \sum_{u=1}^{t} \mu_u = \frac{\gamma}{t} \sum_{u=1}^{t} \sum_{k=1}^{u} (I - \gamma H)^{u-k} \xi_k = \frac{1}{t} \sum_{k=1}^{t} H^{-1} \left( I - (I - \gamma H)^{t-k+1} \right) \xi_k$$

$$\mathbb{E}\left[ \left\| H^{u/2} \bar{\mu}_t \right\|^2 \right] = \frac{1}{t^2} \mathbb{E} \sum_{k=1}^{t} \mathrm{tr} \left[ \left( I - (I - \gamma H)^{t-k+1} \right)^2 H^{u-2} \xi_k \otimes \xi_k \right]$$

$$\leqslant \frac{\sigma^2}{t^2} \sum_{k=1}^{t} \mathrm{tr} \left[ \left( I - (I - \gamma H)^k \right)^2 H^{u-1} \right] \text{ using } \mathbb{E}\left[ \xi_t \otimes \xi_t \right] \preccurlyeq \sigma^2 H.$$

Now, let $(\lambda_i)_{i \in \mathbb{N}^*}$ be the non-increasing sequence of eigenvalues of the operator $H$. We obtain:

$$\mathbb{E}\left[ \left\| H^{u/2} \bar{\mu}_t \right\|^2 \right] \leqslant \frac{\sigma^2}{t^2} \sum_{k=1}^{t} \sum_{i=1}^{\infty} \left( I - (I - \gamma \lambda_i)^k \right)^2 \lambda_i^{u-1}.$$

We can now use a simple result[2] that for any $\rho \in [0, 1]$, $k \geqslant 1$ and $u \in [0, \frac{1}{\alpha} + 1]$, we have : $(1 - (1 - \rho)^k)^2 \leqslant (k\rho)^{1-u+1/\alpha}$, applied to $\rho = \gamma \lambda_i$. We get, by comparing sums to integrals:

$$\mathbb{E}\left[ \left\| H^{u/2} \bar{\mu}_t \right\|^2 \right] \leqslant \frac{\sigma^2}{t^2} \sum_{k=1}^{t} \sum_{i=1}^{\infty} \left( I - (I - \gamma \lambda_i)^k \right)^2 \lambda_i^{u-1}$$

$$\leqslant \frac{\sigma^2}{t^2} \sum_{k=1}^{t} \sum_{i=1}^{\infty} (k\gamma \lambda_i)^{1-u+1/\alpha} \lambda_i^{u-1}$$

$$\leqslant \frac{\sigma^2}{t^2} \gamma^{1-u+1/\alpha} \mathrm{tr} H^{1/\alpha} \sum_{k=1}^{t} k^{1-u+1/\alpha}$$

$$\leqslant \frac{\sigma^2}{t^2} \gamma^{1-u+1/\alpha} \mathrm{tr} H^{1/\alpha} \int_{1}^{t} y^{1-u+1/\alpha} dy$$

$$\leqslant \frac{\sigma^2}{t^2} \gamma^{1-u} \gamma^{1/\alpha} \mathrm{tr} H^{1/\alpha} \frac{t^{2-u+1/\alpha}}{2 - u + 1/\alpha}$$

$$\leqslant \sigma^2 \gamma^{1-u} \gamma^{1/\alpha} \mathrm{tr} H^{1/\alpha} t^{1/\alpha - u},$$

which shows the desired result. $\qquad\qquad\square$

## A.3 Relating the semi-stochastic recursion to the main recursion

Then, to relate the semi-stochastic recursion with the true one, we use an expansion in the powers of $\gamma$ using recursively the perturbation idea from [38].

For $r \geqslant 0$, we define the sequence $(\mu_t^r)_{t \in \mathbb{N}}$, for $t \geqslant 1$,

$$\mu_t^r = (I - \gamma H)\mu_{t-1}^r + \gamma \Xi_t^r, \text{ with } \Xi_t^r = \begin{cases} (H - z_t \otimes z_t)\mu_{t-1}^{r-1} & \text{if } r \geqslant 1 \\ \Xi_t^0 = \xi_t \end{cases} . \tag{4}$$

We will show that $\mu_t \simeq \sum_{i=0}^{\infty} \mu_t^i$. To do so, notice that for $r \geqslant 0$, $\mu_t - \sum_{i=0}^{r} \mu_t^i$ follows the recursion:

$$\mu_t - \sum_{i=0}^{r} \mu_t^i = (I - z_t \otimes z_t) \left( \mu_{t-1} - \sum_{i=0}^{r} \mu_{t-1}^i \right) + \gamma \Xi_t^{r+1}, \tag{5}$$

so that by bounding the covariance operator we can apply a classical SGD result. This is the purpose of the following lemma.

**Lemma 2** (Bound on covariance operator). *For any $r \geqslant 0$, we have the following inequalities:*

$$\mathbb{E}\left[\Xi_t^r \otimes \Xi_t^r\right] \preccurlyeq \gamma^r R^{2r} \sigma^2 H \quad and \quad \mathbb{E}\left[\mu_t^r \otimes \mu_t^r\right] \preccurlyeq \gamma^{r+1} R^{2r} \sigma^2 I. \tag{6}$$

*Proof.* We propose a proof by induction on $r$. For $r = 0$, and $t \geqslant 0$, $\mathbb{E}\left[\Xi_t^0 \otimes \Xi_t^0\right] = \mathbb{E}\left[\xi_t \otimes \xi_t\right] \preccurlyeq \sigma^2 H$ by assumption. Moreover,

$$\mathbb{E}\left[\mu_t^0 \otimes \mu_t^0\right] = \gamma^2 \sum_{k=1}^{t-1} (I - \gamma H)^{t-k} \mathbb{E}\left[\Xi_t^0 \otimes \Xi_t^0\right] (I - \gamma H)^{t-k} \preccurlyeq \gamma^2 \sigma^2 \sum_{k=1}^{t-1} (I - \gamma H)^{2(t-k)} H \preccurlyeq \gamma \sigma^2 I.$$

Then, for $r \geqslant 1$,

$$\begin{aligned}
\mathbb{E}\left[\Xi_t^{r+1} \otimes \Xi_t^{r+1}\right] &\preccurlyeq \mathbb{E}[(H - z_t \otimes z_t)\mu_{t-1}^r \otimes \mu_{t-1}^r (H - z_t \otimes z_t)] \\
&= \mathbb{E}[(H - z_t \otimes z_t)\mathbb{E}[\mu_{t-1}^r \otimes \mu_{t-1}^r](H - z_t \otimes z_t)] \\
&\preccurlyeq \gamma^{r+1} R^{2r} \sigma^2 \mathbb{E}[(H - z_t \otimes z_t)^2] \\
&\preccurlyeq \gamma^{r+1} R^{2r+2} \sigma^2 H.
\end{aligned}$$

And,

$$\begin{aligned}
\mathbb{E}\left[\mu_t^{r+1} \otimes \mu_t^{r+1}\right] &= \gamma^2 \sum_{k=1}^{t-1} (I - \gamma H)^{t-k} \mathbb{E}\left[\Xi_t^{r+1} \otimes \Xi_t^{r+1}\right] (I - \gamma H)^{t-k} \\
&\preccurlyeq \gamma^{r+3} R^{2r+2} \sigma^2 \sum_{k=1}^{t-1} (I - \gamma H)^{2(t-k)} H \preccurlyeq \gamma^{r+2} R^{2r+2} \sigma^2 I,
\end{aligned}$$

which thus shows the lemma by induction. $\qquad\square$

To bound $\mu_t - \sum_{i=0}^r \mu_t^i$, we prove a very loose result for the average iterate, that will be sufficient for our purpose.

**Lemma 3** (Bounding SGD recursion). *Let us consider the following recursion $\mu_t = (I - \gamma z_t \otimes z_t)\mu_{t-1} + \gamma \xi_t$ starting at $\mu_0 = 0$. Assume that $\mathbb{E}[z_t \otimes z_t] = H$, $\mathbb{E}\left[\xi_t\right] = 0$, $\|x_t\|^2 \leqslant R^2$, $\mathbb{E}\left[\xi_t \otimes \xi_t\right] \preccurlyeq \sigma^2 H$ and $\gamma R^2 < I$, then for $u \in [0, \frac{1}{\alpha} + 1]$:*

$$\mathbb{E}\left[\left\|H^{u/2} \bar{\mu}_t\right\|^2\right] \leqslant \sigma^2 \gamma^2 R^u \mathrm{tr} H\, t. \tag{7}$$

*Proof.* Let us define the operators for $j \leqslant i$: $M_j^i = (I - \gamma z_{i(i)} \otimes z_{i(i)}) \cdots (I - \gamma z_{i(j)} \otimes z_{i(j)})$ and $M_{i+1}^i = I$. Since $\mu_0 = 0$, note that we have we have, $\mu_i = \gamma \sum_{k=1}^i M_{k+1}^i \xi_k$. Hence, for $i \geqslant 1$,

$$\begin{aligned}
\mathbb{E}\left\|H^{u/2} \mu_i\right\|^2 &= \gamma^2 \mathbb{E} \sum_{k,j} \langle M_{j+1}^i \xi_j, H^u M_{k+1}^i \xi_k \rangle \\
&= \gamma^2 \mathbb{E} \sum_{k=1}^i \langle M_{k+1}^i \xi_k, H^u M_{k+1}^i \xi_k \rangle \\
&= \gamma^2 \mathrm{tr}\left(\mathbb{E}\left[\sum_{k=1}^i {M_{k+1}^i}^* H^u M_{k+1}^i \xi_k \otimes \xi_k\right]\right) \leqslant \sigma^2 \gamma^2 \mathbb{E}\left[\sum_{k=1}^i \mathrm{tr}\left({M_{k+1}^i}^* H^u M_{k+1}^i H\right)\right] \\
&\leqslant \sigma^2 \gamma^2 R^u i\, \mathrm{tr} H,
\end{aligned}$$

because $\mathrm{tr}\left({M_{k+1}^i}^* H^u M_{k+1}^i H\right) \leqslant R^u \mathrm{tr} H$. Then,

$$\begin{aligned}
\mathbb{E}\left\|H^{u/2} \bar{\mu}_t\right\|^2 &= \frac{1}{t^2} \sum_{i,j} \langle H^{u/2} \mu_i, H^{u/2} \mu_j \rangle \\
&\leqslant \frac{1}{t^2} \mathbb{E}\left(\sum_{i=1}^t \left\|H^{u/2} \mu_i\right\|\right)^2 \leqslant \frac{1}{t} \sum_{i=1}^t \mathbb{E}\left\|H^{u/2} \mu_i\right\|^2 \leqslant \sigma^2 \gamma^2 R^u \mathrm{tr} H\, t,
\end{aligned}$$

which finishes the proof of Lemma 3. $\qquad\square$

### A.4 Final steps of the proof

We have now all the material to conclude. Indeed by the triangular inequality:

$$\left(\mathbb{E}\left\|H^{u/2}\bar{\mu}_t\right\|^2\right)^{1/2} \leqslant \sum_{i=1}^{r}\underbrace{\left(\mathbb{E}\left\|H^{u/2}\bar{\mu}_t^i\right\|^2\right)}_{\text{Lemma 1}}^{1/2} + \underbrace{\left(\mathbb{E}\left\|H^{u/2}\left(\bar{\mu}_t - \sum_{i=1}^{r}\bar{\mu}_t^i\right)\right\|^2\right)}_{\text{Lemma 3}}^{1/2}.$$

With Lemma 2, we have all the bounds on the covariance of the noise, so that:

$$\left(\mathbb{E}\left\|H^{u/2}\bar{\mu}_t\right\|^2\right)^{1/2} \leqslant \sum_{i=1}^{r}\left(\gamma^i R^{2i}\sigma^2\gamma^{1-u}\,\gamma^{1/\alpha}\mathrm{tr}H^{1/\alpha}t^{1/\alpha-u}\right)^{1/2} + \left(\gamma^{r+2}R^{2r+u}\mathrm{tr}H\,t\right)^{1/2}$$

$$\leqslant (\sigma^2\gamma^{1-u}\,\gamma^{1/\alpha}\mathrm{tr}H^{1/\alpha}t^{1/\alpha-u})^{1/2}\sum_{i=1}^{r}\left(\gamma R^2\right)^{i/2} + \left(\gamma^{r+2}R^{2r+u}\mathrm{tr}H\,t\right)^{1/2}.$$

Now we make $r$ go to infinity and we obtain:

$$\left(\mathbb{E}\left\|H^{u/2}\bar{\mu}_t\right\|^2\right)^{1/2} \leqslant (\sigma^2\gamma^{1-u}\,\gamma^{1/\alpha}\mathrm{tr}H^{1/\alpha}t^{1/\alpha-u})^{1/2}\frac{1}{1-\sqrt{\gamma R^2}} + \underbrace{\left(\gamma^{r+2}R^{2r+u}\mathrm{tr}H\,t\right)^{1/2}}_{\underset{r\to\infty}{\longrightarrow}0}$$

Hence with $\gamma R^2 \leqslant 1/4$,

$$\mathbb{E}\left\|H^{u/2}\bar{\mu}_t\right\|^2 \leqslant 4\sigma^2\gamma^{1-u}\,\gamma^{1/\alpha}\mathrm{tr}H^{1/\alpha}t^{1/\alpha-u},$$

which finishes to prove Proposition 1.

## B  Proof sketch for Theorem 1

We consider the batch gradient descent recursion, started from $\eta_0 = 0$, with the same step-size:

$$\eta_t = \eta_{t-1} + \frac{\gamma}{n}\sum_{i=1}^{n}\left(y_i - \langle\eta_{t-1},\Phi(x_i)\rangle_{\mathcal{H}}\right)\Phi(x_i),$$

as well as its averaged version $\bar{\eta}_t = \frac{1}{t}\sum_{i=0}^{t}\eta_i$. We obtain a recursion for $\theta_t - \eta_t$, with the initialization $\theta_0 - \eta_0 = 0$, as follows:

$$\theta_t - \eta_t = \left[I - \Phi(x_{i(u)})\otimes_{\mathcal{H}}\Phi(x_{i(u)})\right](\theta_{t-1} - \eta_{t-1}) + \gamma\xi_t^1 + \gamma\xi_t^2,$$

with $\xi_t^1 = y_{i(u)}\Phi(x_{i(u)}) - \frac{1}{n}\sum_{i=1}^{n}y_i\Phi(x_i)$ and $\xi_t^2 = \left[\Phi(x_{i(u)})\otimes_{\mathcal{H}}\Phi(x_{i(u)}) - \frac{1}{n}\sum_{i=1}^{n}\Phi(x_i)\otimes_{\mathcal{H}}\Phi(x_i)\right]\eta_{t-1}$. We decompose the performance $F(\theta_t)$ in two parts, one analyzing the performance of batch gradient descent, one analyzing the deviation $\theta_t - \eta_t$, using

$$\mathbb{E}F(\bar{\theta}_t) - F(\theta_*) \leqslant 2\mathbb{E}\left[\|\Sigma^{1/2}(\theta_t - \eta_t)\|_{\mathcal{H}}^2\right] + 2\left[\mathbb{E}F(\bar{\eta}_t) - F(\theta_*)\right].$$

We denote by $\hat{\Sigma}_n = \frac{1}{n}\sum_{i=1}^{n}\Phi(x_i)\otimes\Phi(x_i)$ the empirical second-order moment.

**Deviation $\theta_t - \eta_t$.** Denoting by $\mathcal{G}$ the $\sigma$-field generated by the data and by $\mathcal{F}_t$ the $\sigma$-field generated by $i(1),\dots,i(t)$, then, we have $\mathbb{E}(\xi_t^1|\mathcal{G},\mathcal{F}_{t-1}) = \mathbb{E}(\xi_t^2|\mathcal{G},\mathcal{F}_{t-1}) = 0$, thus we can apply results for averaged SGD (see Proposition 1 of the Appendix) to get the following lemma.

**Lemma 4.** *For any $t \geqslant 1$, if $\mathbb{E}\left[(\xi_t^1 + \xi_t^2)\otimes_{\mathcal{H}}(\xi_t^1 + \xi_t^2)|\mathcal{G}\right] \preccurlyeq \tau^2\hat{\Sigma}_n$, and $4\gamma R^2 = 1$, under Assumptions (A1), (A2), (A4),*

$$\mathbb{E}\left[\|\hat{\Sigma}_n^{1/2}(\bar{\theta}_t - \bar{\eta}_t)\|_{\mathcal{H}}^2|\mathcal{G}\right] \leqslant \frac{8\tau^2\gamma^{1/\alpha}\mathrm{tr}\,\hat{\Sigma}_n^{1/\alpha}}{t^{1-1/\alpha}}. \tag{8}$$

In order to obtain the bound, we need to bound $\tau^2$ (which is dependent on $\mathcal{G}$) and go from a bound with the empirical covariance matrix $\hat{\Sigma}_n$ to bounds with the population covariance matrix $\Sigma$.

We have

$$\mathbb{E}\big[\xi_t^1 \otimes_{\mathcal{H}} \xi_t^1 | \mathcal{G}\big] \preccurlyeq_{\mathcal{H}} \mathbb{E}\big[y_{i(u)}^2 \Phi(x_{i(u)}) \otimes_{\mathcal{H}} \Phi(x_{i(u)}) | \mathcal{G}\big] \preccurlyeq_{\mathcal{H}} \|y\|_\infty^2 \hat{\Sigma}_n \preccurlyeq_{\mathcal{H}} (\sigma + \sup_{x \in \mathcal{X}} \langle \theta_*, \Phi(x) \rangle_{\mathcal{H}})^2 \hat{\Sigma}_n$$

$$\mathbb{E}\big[\xi_t^2 \otimes \xi_t^2 | \mathcal{G}\big] \preccurlyeq_{\mathcal{H}} \mathbb{E}\big[\langle \eta_{t-1}, \Phi(x_{i(u)}) \rangle^2 \Phi(x_{i(u)}) \otimes_{\mathcal{H}} \Phi(x_{i(u)}) | \mathcal{G}\big] \preccurlyeq_{\mathcal{H}} \sup_{t \in \{0,...,T-1\}} \sup_{x \in \mathcal{X}} \langle \eta_t, \Phi(x) \rangle_{\mathcal{H}}^2 \hat{\Sigma}_n$$

Therefore $\tau^2 = 2M^2 + 2 \sup_{t \in \{0,...,T-1\}} \sup_{x \in \mathcal{X}} \langle \eta_t, \Phi(x) \rangle_{\mathcal{H}}^2$ or using Assumption (A3) $\tau^2 = 2M^2 + 2 \sup_{t \in \{0,...,T-1\}} R^{2\mu} \kappa_\mu^2 \|\Sigma^{1/2-\mu/2} \eta_t\|_{\mathcal{H}}^2$.

In the proof, we rely on an event (that depend on $\mathcal{G}$) where $\hat{\Sigma}_n$ is close to $\Sigma$. This leads to the the following Lemma that bounds the deviation $\bar{\theta}_t - \bar{\eta}_t$.

**Lemma 5.** *For any $t \geqslant 1$, $4\gamma R^2 = 1$, under Assumptions (A1), (A2), (A4),*

$$\mathbb{E}\big[\|\Sigma^{1/2}(\bar{\theta}_t - \bar{\eta}_t)\|_{\mathcal{H}}^2\big] \leqslant 16\tau_\infty^2 \left[ R^{-2/\alpha} \operatorname{tr} \Sigma^{1/\alpha} t^{1/\alpha} \left( \frac{1}{t} + \left( \frac{4}{\mu} \frac{\log n}{n} \right)^{1/\mu} \right) + 1 \right]. \tag{9}$$

We make the following remark on the bound.

**Remark 1.** *Note that as defined in the proof $\tau_\infty$ may diverge in some cases as*

$$\tau_\infty^2 = \begin{cases} O(1) & \text{when } \mu \leqslant 2r, \\ O(n^{\mu - 2r}) & \text{when } 2r \leqslant \mu \leqslant 2r + 1/\alpha, \\ O(n^{1 - 2r/\mu}) & \text{when } \mu \geqslant 2r + 1/\alpha, \end{cases}$$

*with $O(\cdot)$ are defined explicitly in the proof.*

**Convergence of batch gradient descent.** The main result is summed up in the following lemma, with $t = O(n^{1/\mu})$ and $t \geqslant n$.

**Lemma 6.** *Let $t > 1$, under Assumptions (A1), (A2), (A3), (A4), (A5), (A6), when, with $4\gamma R^2 = 1$,*

$$t = \begin{cases} \Theta(n^{\alpha/(2r\alpha+1)}) & 2r\alpha + 1 > \mu\alpha \\ \Theta(n^{1/\mu} (\log n)^{\frac{1}{\mu}}) & 2r\alpha + 1 \leqslant \mu\alpha. \end{cases} \tag{10}$$

*then,*

$$\mathbb{E}F(\bar{\eta}_t) - F(\theta_*) \leqslant \begin{cases} O(n^{-2r\alpha/(2r\alpha+1)}) & 2r\alpha + 1 > \mu\alpha \\ O(n^{-2r/\mu}) & 2r\alpha + 1 \leqslant \mu\alpha \end{cases} \tag{11}$$

*with $O(\cdot)$ are defined explicitly in the proof.*

**Remark 2.** *In all cases, we can notice that the speed of convergence of Lemma 6 are slower that the ones in Lemma 5, hence, the convergence of the gradient descent controls the rates of convergence of the algorithm.*

# C  Bounding the deviation between SGD and batch gradient descent

In this section, following the proof sketch from Section B, we provide a bound on the deviation $\theta_t - \eta_t$. In all the following let us denote $\mu_t = \theta_t - \eta_t$ that deviation between the stochastic gradient descent recursion and the batch gradient descent recursion.

## C.1  Proof of Lemma 5

We need to (a) go from $\hat{\Sigma}_n$ to $\Sigma$ in the result of Lemma 4 and (b) to have a bound on $\tau$. To prove this result we are going to need the two following lemmas:

**Lemma 7.** *Let $\lambda > 0$, $\delta \in (0,1]$. Under Assumption (A3), when $n \geqslant 11(1 + \kappa_\mu^2 R^{2\mu} \gamma^\mu t^\mu) \log \frac{8R^2}{\lambda\delta}$, the following holds with probability $1 - \delta$,*

$$\left\| (\Sigma + \lambda I)^{1/2}(\hat{\Sigma}_n + \lambda I)^{-1/2} \right\|^2 \leqslant 2. \tag{12}$$

*Proof.* This Lemma is proven and stated lately in Lemma 14 in Section D.3. We recalled it here for the sake of clarity. $\qquad\square$

**Lemma 8.** *Let $\lambda > 0$, $\delta \in (0,1]$. Under Assumption (A3), for $t = O\left(\frac{1}{n^{1/\mu}}\right)$ then the following holds with probability $1 - \delta$,*

$$\tau^2 \leqslant \tau_\infty^2 \quad and \quad \tau_\infty^2 = \begin{cases} O(1), \text{ when } \mu \leqslant 2r, \\ O\left(n^{\mu - 2r}\right), \text{ when } 2r \leqslant \mu \leqslant 2r + 1/\alpha, \\ O\left(n^{1 - 2r/\mu}\right) \text{ when } \mu \geqslant 2r + 1/\alpha, \end{cases} \tag{13}$$

*where the $O(\cdot)$-notation depend only on the parameters of the problem (and is independent of $n$ and $t$).*

*Proof.* This Lemma is a direct implication of Corollary 2 in Section D.3. We recalled it here for the sake of clarity. $\qquad\square$

Note that we can take $\lambda_n^\delta = \left(\frac{\log \frac{n}{\delta}}{n}\right)^{1/\mu}$ so that Lemma 7 result holds. Now we are ready to prove Lemma 5.

*Proof of Lemma 5.* Let $A_{\delta_a}$ be the set for which inequality (12) holds and let $B_{\delta_b}$ be the set for which inequality (13) holds. Note that $\mathbb{P}(A_{\delta_a}^c) = \delta_a$ and $\mathbb{P}(B_{\delta_b}^c) = \delta_b$. We use the following decomposition:

$$\mathbb{E}\left\|\Sigma^{1/2}\bar{\mu}_t\right\|^2 \leqslant \mathbb{E}\left[\left\|\Sigma^{1/2}\bar{\mu}_t\right\|^2 \mathbf{1}_{A_{\delta_a} \cap B_{\delta_b}}\right] + \mathbb{E}\left[\left\|\Sigma^{1/2}\bar{\mu}_t\right\|^2 \mathbf{1}_{A_{\delta_a}^c}\right] + \mathbb{E}\left[\left\|\Sigma^{1/2}\bar{\mu}_t\right\|^2 \mathbf{1}_{B_{\delta_b}^c}\right].$$

First, let us bound roughly $\|\bar{\mu}_t\|^2$.

First, for $i \geqslant 1$, $\|\mu_i\|^2 \leqslant \gamma^2\left(\sum_{i=1}^t \|\xi_i^1\| + \|\xi_i^2\|\right)^2 \leqslant 16R^2\gamma^2\tau^2 t^2$, so that $\|\bar{\mu}_t\|^2 \leqslant \frac{1}{t}\sum_{i=1}^t \|\mu_i\|^2 \leqslant 16R^2\gamma^2\tau^2 t^2$. We can bound similarly $\tau^2 \leqslant 4M^2\gamma^2 R^4 t^2$, so that $\|\bar{\mu}_t\|^2 \leqslant 64R^2 M^2 \gamma^4 t^4$. Thus, for the second term:

$$\mathbb{E}\left[\left\|\Sigma^{1/2}\bar{\mu}_t\right\|^2 \mathbf{1}_{A_{\delta_a}^c}\right] \leqslant 64R^8 M^2 \gamma^4 t^4 \mathbb{E}\mathbf{1}_{A_{\delta_a}^c} \leqslant 64R^8 M^2 \gamma^4 t^4 \delta_a,$$

and for the third term:

$$\mathbb{E}\left[\left\|\Sigma^{1/2}\bar{\mu}_t\right\|^2 \mathbf{1}_{B_{\delta_b}^c}\right] \leqslant 64R^8 M^2 \gamma^4 t^4 \mathbb{E}\mathbf{1}_{B_{\delta_b}^c} \leqslant 64R^8 M^2 \gamma^4 t^4 \delta_b.$$

And on for the first term,

$$\mathbb{E}\left[\left\|\Sigma^{1/2}\bar{\mu}_t\right\|^2 \mathbf{1}_{A_{\delta_a} \cap B_{\delta_b}}\right] \leqslant \mathbb{E}\left[\left\|\Sigma^{1/2}(\Sigma + \lambda_n^\delta I)^{-1/2}\right\|^2 \left\|(\Sigma + \lambda_n^\delta I)^{1/2}(\hat{\Sigma}_n + \lambda_n^\delta I)^{-1/2}\right\|^2\right.$$

$$\left. \left\|(\hat{\Sigma}_n + \lambda_n^\delta I)^{1/2}\bar{\mu}_t\right\|^2 \mathbf{1}_{A_{\delta_a} \cap B_{\delta_b}} \mid \mathcal{G}\right]$$

$$\leqslant 2\mathbb{E}\left[\left\|(\hat{\Sigma}_n + \lambda_n^\delta I)^{1/2}\bar{\mu}_t\right\|^2 \mid \mathcal{G}\right]$$

$$= 2\mathbb{E}\left[\left\|\hat{\Sigma}_n^{1/2}\bar{\mu}_t\right\|^2 \mid \mathcal{G}\right] + 2\lambda_n^\delta \mathbb{E}\left[\|\bar{\mu}_t\|^2 \mid \mathcal{G}\right]$$

$$\leqslant 16\tau_\infty^2 \frac{\gamma^{1/\alpha}\mathbb{E}\left[\operatorname{tr} \hat{\Sigma}_n^{1/\alpha}\right]}{t^{1 - 1/\alpha}} + 8\lambda_n^\delta \tau_\infty^2 \gamma^{1/\alpha}\mathbb{E}\left[\operatorname{tr} \hat{\Sigma}_n^{1/\alpha}\right] t^{1/\alpha},$$

using Proposition 1 twice with $u = 1$ for the left term and $u = 1$ for the right one.

As $x \to x^{1/\alpha}$ is a concave function, we can apply Jensen's inequality to have :

$$\mathbb{E}\left[\operatorname{tr}(\hat{\Sigma}_n^{1/\alpha})\right] \leqslant \operatorname{tr}\Sigma^{1/\alpha},$$

so that:

$$\mathbb{E}\left[\left\|\Sigma^{1/2}\bar{\mu}_t\right\|^2 \mathbf{1}_{A_{\delta_a}\cap B_{\delta_b}}\right] \leqslant 16\tau_\infty^2 \frac{\gamma^{1/\alpha}\mathrm{tr}\,\Sigma^{1/\alpha}}{t^{1-1/\alpha}} + 8\lambda_n^\delta\tau_\infty^2\gamma\,\gamma^{1/\alpha}\mathrm{tr}\,\Sigma^{1/\alpha}t^{1/\alpha}$$

$$\leqslant 16\tau_\infty^2\gamma^{1/\alpha}\mathrm{tr}\,\Sigma^{1/\alpha}t^{1/\alpha}\left(\frac{1}{t} + \lambda_n^\delta\right).$$

Now, we take $\delta_a = \delta_b = \frac{\tau_\infty^2}{4M^2R^8\gamma^4t^4}$ and this concludes the proof of Lemma 5, with the bound:

$$\mathbb{E}\left\|\Sigma^{1/2}\bar{\mu}_t\right\|^2 \leqslant 16\tau_\infty^2\gamma^{1/\alpha}\mathrm{tr}\,\Sigma^{1/\alpha}t^{1/\alpha}\left(\frac{1}{t} + \left(\frac{2 + 2\log M + 4\log(\gamma R^2) + 4\log t}{n}\right)^{1/\mu}\right).$$

$\square$

# D   Convergence of batch gradient descent

In this section we prove the convergence of averaged batch gradient descent to the target function. In particular, since the proof technique is valid for the wider class of algorithms known as spectral filters [15, 14], we will do the proof for a generic spectral filter (in Lemma 9, Sect. D.1 we prove that averaged batch gradient descent is a spectral filter).

In Section D.1 we provide the required notation and additional definitions. In Section D.2, in particular in Theorem D.2 we perform an analytical decomposition of the excess risk of the averaged batch gradient descent, in terms of basic quantities that will be controlled in expectation (or probability) in the next sections. In Section D.3 the various quantites obtained by the analytical decomposition are controlled, in particular, Corollary 2 controls the $L^\infty$ norm of the averaged batch gradient descent algorithm. Finally in Section D.4, the main result, Theorem 3 controlling in expectation of the excess risk of the averaged batch gradient descent estimator is provided. In Corollary 3, a version of the result of Theorem 3 is given, with explicit rates for the regularization parameters and of the excess risk.

## D.1   Notations

In this subsection, we study the convergence of batch gradient descent. For the sake of clarity we consider the RKHS framework (which includes the finite-dimensional case). We will thus consider elements of $\mathcal{H}$ that are naturally embedded in $L_2(d\rho_\mathcal{X})$ by the operator $S$ from $\mathcal{H}$ to $L_2(d\rho_\mathcal{X})$ and such that: $(Sg)(x) = \langle g, K_x\rangle$, where we have $\Phi(x) = K_x = K(\cdot, x)$ where $K : \mathcal{X} \to \mathcal{X} \to \mathbb{R}$ is the kernel. We recall the recursion for $\eta_t$ in the case of an RKHS feature space with kernel $K$:

$$\eta_t = \eta_{t-1} + \frac{\gamma}{n}\sum_{i=1}^n \left(y_i - \langle\eta_{t-1}, K_{x_i}\rangle_\mathcal{H}\right)K_{x_i},$$

Let us begin with some notations. In the following we will often use the letter $g$ to denote vectors of $\mathcal{H}$, hence, $Sg$ will denote functions of $L_2(d\rho_\mathcal{X})$. We also define the following operators (we may also use their adjoints, denoted with a $*$):

- The operator $\hat{S}_n$ from $\mathcal{H}$ to $\mathbb{R}^n$, $\hat{S}_n g = \frac{1}{\sqrt{n}}(g(x_1),\ldots g(x_n))$.

- The operators from $\mathcal{H}$ to $\mathcal{H}$, $\Sigma$ and $\hat{\Sigma}_n$, defined respectively as $\Sigma = \mathbb{E}\left[K_x \otimes K_x\right] = \int_\mathcal{X} K_x \otimes K_x d\rho_\mathcal{X}$ and $\hat{\Sigma}_n = \frac{1}{n}\sum_{i=1}^n K_{x_i} \otimes K_{x_i}$. Note that $\Sigma$ is the covariance operator.

- The operator $\mathcal{L} : L_2(d\rho_\mathcal{X}) \to L_2(d\rho_\mathcal{X})$ is defined by

$$(\mathcal{L}f)(x) = \int_\mathcal{X} K(x, z)f(z)d\rho_\mathcal{X}(x), \quad \forall f \in L_2(d\rho_\mathcal{X}).$$

Moreover denote by $\mathcal{N}(\lambda)$ the so called *effective dimension* of the learning problem, that is defined as

$$\mathcal{N}(\lambda) = \mathrm{tr}(\mathcal{L}(\mathcal{L} + \lambda I)^{-1}),$$

for $\lambda > 0$. Recall that by Assumption (A4), there exists $\alpha \geqslant 1$ and $Q > 0$ such that

$$\mathcal{N}(\lambda) \leqslant Q\lambda^{-1/\alpha}, \quad \forall\lambda > 0.$$

We can take $Q = \mathrm{tr}\Sigma^{1/\alpha}$.

- $P : L_2(d\rho_{\mathcal{X}}) \to L_2(d\rho_{\mathcal{X}})$ projection operator on $\mathcal{H}$ for the $L_2(d\rho_{\mathcal{X}})$ norm s.t. $\mathrm{ran}P = \mathrm{ran}S$.

Denote by $f_\rho$ the function so that $f_\rho(x) = \mathbb{E}[y|x] \in L_2(d\rho_{\mathcal{X}})$ the minimizer of the expected risk, defined by $F(f) = \int_{X \times \mathbb{R}}(f(x) - y)^2 d\rho(x,y)$.

**Remark 3** (On Assumption (A5))**.** *With the notation above, we express assumption (A5), more formally, w.r.t. Hilbert spaces with infinite dimensions, as follows. There exists $r \in [0, 1]$ and $\phi \in L_2(d\rho_{\mathcal{X}})$, such that*

$$Pf_\rho = \mathcal{L}^r \phi.$$

**(A6)** *Let $q \in [1, \infty]$ be such that $\|f_\rho - Pf_\rho\|_{L^{2q}(\mathcal{X}, \rho_{\mathcal{X}})} < \infty$.*

The assumption above is always true for $q = 1$, moreover when the kernel is universal it is true even for $q = \infty$. Moreover if $r \geqslant 1/2$ then it is true for $q = \infty$. Note that we make the calculation in this Appendix for a general $q \in [1, \infty]$, but we presented the results for $q = \infty$ in the main paper. The following proposition relates the excess risk to a certain norm.

**Proposition 2.** *When $\widehat{g} \in \mathcal{H}$,*

$$F(\widehat{g}) - \inf_{g \in \mathcal{H}} F(g) = \|S\widehat{g} - Pf_\rho\|^2_{L_2(d\rho_{\mathcal{X}})}.$$

We introduce the following function $g_\lambda \in \mathcal{H}$ that will be useful in the rest of the paper $g_\lambda = (\Sigma + \lambda I)^{-1}S^* f_\rho$.

We introduce the estimators of the form, for $\lambda > 0$,

$$\widehat{g}_\lambda = q_\lambda(\hat{\Sigma}_n)\hat{S}_n^* \hat{y},$$

where $q_\lambda : \mathbb{R}_+ \to \mathbb{R}_+$ is a function called *filter*, that essentially approximates $x^{-1}$ with the approximation controlled by $\lambda$. Denote moreover with $r_\lambda$ the function $r_\lambda(x) = 1 - xq_\lambda(x)$. The following definition precises the form of the filters we want to analyze. We then prove in Lemma 9 that our estimator corresponds to such a filter.

**Definition 1** (Spectral filters)**.** *Let $q_\lambda : \mathbb{R}_+ \to \mathbb{R}_+$ be a function parametrized by $\lambda > 0$. $q_\lambda$ is called a* filter *when there exists $c_q > 0$ for which*

$$\lambda q_\lambda(x) \leqslant c_q, \quad r_\lambda(x)x^u \leqslant c_q\lambda^u, \quad \forall x > 0, \lambda > 0, u \in [0,1].$$

We now justify that we study estimators of the form $\widehat{g}_\lambda = q_\lambda(\hat{\Sigma}_n)\hat{S}_n^* \hat{y}$ with the following lemma. Indeed, we show that the average of batch gradient descent can be represented as a filter estimator, $\widehat{g}_\lambda$, for $\lambda = 1/(\gamma t)$.

**Lemma 9.** *For $t > 1$, $\lambda = 1/(\gamma t)$, $\bar{\eta}_t = \widehat{g}_\lambda$, with respect to the filter, $q^\eta(x) = \left(1 - \frac{1-(1-\gamma x)^t}{\gamma t x}\right)\frac{1}{x}$.*

*Proof.* Indeed, for $t > 1$,

$$\eta_t = \eta_{t-1} + \frac{\gamma}{n}\sum_{i=1}^n \left(y_i - \langle \eta_{t-1}, K_{x_i}\rangle_{\mathcal{H}}\right)K_{x_i}$$

$$= \eta_{t-1} + \gamma(\hat{S}_n^* \hat{y} - \hat{\Sigma}_n \eta_{t-1})$$

$$= (I - \gamma\hat{\Sigma}_n)\eta_{t-1} + \gamma\hat{S}_n^* \hat{y}$$

$$= \gamma\sum_{k=0}^{t-1}(I - \gamma\hat{\Sigma}_n)^k \hat{S}_n^* \hat{y} = \left[I - (I - \gamma\hat{\Sigma}_n)^t\right]\hat{\Sigma}_n^{-1}\hat{S}_n^* \hat{y},$$

leading to

$$\bar{\eta}_t = \frac{1}{t}\sum_{i=0}^t \eta_i = q^\eta\left(\hat{\Sigma}_n\right)\hat{S}_n^* \hat{y}.$$

Now, we prove that $q$ has the properties of a filter. First, for $t > 1$, $\frac{1}{\gamma t}q^\eta(x) = \left(1 - \frac{1-(1-\gamma x)^t}{\gamma t x}\right)\frac{1}{\gamma t x}$ is a decreasing function so that $\frac{1}{\gamma t}q^\eta(x) \leqslant \frac{1}{\gamma t}q^\eta(0) \leqslant 1$. Second for $u \in [0,1]$, $x^u(1 - xq^\eta(x)) = \frac{1-(1-\gamma x)^t}{\gamma t x}x^u$. As used in Section A.2, $1 - (1 - \gamma x)^t \leqslant (\gamma t x)^{1-u}$, so that, $r^\eta(x)x^u \leqslant \frac{(\gamma t x)^{1-u}}{\gamma t x}x^u = \frac{1}{(\gamma t)^u}$, this concludes the proof that $q^\eta$ is indeed a filter. $\square$

## D.2 Analytical decomposition

**Lemma 10.** *Let $\lambda > 0$ and $s \in (0, 1/2]$. Under Assumption (A5) (see Rem. 3), the following holds*

$$\|\mathcal{L}^{-s} S(\widehat{g}_\lambda - g_\lambda)\|_{L_2(d\rho_{\mathcal{X}})} \leqslant 2\lambda^{-s}\beta^2 c_q \|\Sigma_\lambda^{-1/2}(\hat{S}_n^* \hat{y} - \hat{\Sigma}_n g_\lambda)\|_{\mathcal{H}} + 2\beta c_q \|\phi\|_{L_2(d\rho_{\mathcal{X}})} \lambda^{r-s},$$

*where $\beta := \|\Sigma_\lambda^{1/2} \widehat{\Sigma}_{n\lambda}^{-1/2}\|$.*

*Proof.* By Prop. 2, we can characterize the excess risk of $\widehat{g}_\lambda$ in terms of the $L_2(d\rho_{\mathcal{X}})$ squared norm of $S\widehat{g}_\lambda - Pf_\rho$. In this paper, simplifying the analysis of [14], we perform the following decomposition

$$\mathcal{L}^{-s} S(\widehat{g}_\lambda - g_\lambda) = \mathcal{L}^{-s} S\widehat{g}_\lambda - \mathcal{L}^{-s} Sq_\lambda(\hat{\Sigma}_n)\hat{\Sigma}_n g_\lambda$$
$$+ \; \mathcal{L}^{-s} Sq_\lambda(\hat{\Sigma}_n)\hat{\Sigma}_n g_\lambda - \mathcal{L}^{-s} Sg_\lambda.$$

**Upper bound for the first term.** By using the definition of $\widehat{g}_\lambda$ and multiplying and dividing by $\Sigma_\lambda^{1/2}$, we have that

$$\mathcal{L}^{-s} S\widehat{g}_\lambda - \mathcal{L}^{-s} Sq_\lambda(\hat{\Sigma}_n)\hat{\Sigma}_n g_\lambda = \mathcal{L}^{-s} Sq_\lambda(\hat{\Sigma}_n)(\hat{S}_n^* \hat{y} - \hat{\Sigma}_n g_\lambda)$$
$$= \mathcal{L}^{-s} Sq_\lambda(\hat{\Sigma}_n)\Sigma_\lambda^{1/2} \, \Sigma_\lambda^{-1/2}(\hat{S}_n^* \hat{y} - \hat{\Sigma}_n g_\lambda),$$

from which

$$\|\mathcal{L}^{-s} S(\widehat{g}_\lambda - q_\lambda(\hat{\Sigma}_n)\hat{\Sigma}_n g_\lambda)\|_{L_2(d\rho_{\mathcal{X}})} \leqslant \|\mathcal{L}^{-s} Sq_\lambda(\hat{\Sigma}_n)\Sigma_\lambda^{1/2}\| \, \|\Sigma_\lambda^{-1/2}(\hat{S}_n^* \hat{y} - \hat{\Sigma}_n g_\lambda)\|_{\mathcal{H}}.$$

**Upper bound for the second term.** By definition of $r_\lambda(x) = 1 - xq_\lambda(x)$ and $g_\lambda = \Sigma_\lambda^{-1} S^* f_\rho$,

$$\mathcal{L}^{-s} Sq_\lambda(\hat{\Sigma}_n)\hat{\Sigma}_n g_\lambda - \mathcal{L}^{-s} Sg_\lambda = \mathcal{L}^{-s} S(q_\lambda(\hat{\Sigma}_n)\hat{\Sigma}_n - I)g_\lambda$$
$$= -\mathcal{L}^{-s} Sr_\lambda(\hat{\Sigma}_n) \, \Sigma_\lambda^{-(1/2-r)} \, \Sigma_\lambda^{-1/2-r} S^* \mathcal{L}^r \, \phi,$$

where in the last step we used the fact that $S^* f_\rho = S^* Pf_\rho = S^* \mathcal{L}^r \phi$, by Asm. (A5) (see Rem. 3). Then

$$\|\mathcal{L}^{-s} S(q_\lambda(\hat{\Sigma}_n)\hat{\Sigma}_n - I)g_\lambda)\|_{L_2(d\rho_{\mathcal{X}})} \leqslant \|\mathcal{L}^{-s} Sr_\lambda(\hat{\Sigma}_n)\| \|\Sigma_\lambda^{-(1/2-r)}\| \|\Sigma_\lambda^{-1/2-r} S^* \mathcal{L}^r\| \|\phi\|_{L_2(d\rho_{\mathcal{X}})}$$
$$\leqslant \lambda^{-(1/2-r)} \|\mathcal{L}^{-s} Sr_\lambda(\hat{\Sigma}_n)\| \|\phi\|_{L_2(d\rho_{\mathcal{X}})},$$

where the last step is due to the fact that $\|\Sigma_\lambda^{-(1/2-r)}\| \leqslant \lambda^{-(1/2-r)}$ and that $S^* \mathcal{L}^{2r} S = S^*(SS^*)^{2r} S = (S^* S)^{2r} S^* S = \Sigma^{1+2r}$ from which

$$\|\Sigma_\lambda^{-1/2-r} S^* \mathcal{L}^r\|^2 = \|\Sigma_\lambda^{-1/2-r} S^* \mathcal{L}^{2r} S\Sigma_\lambda^{-1/2-r}\| = \|\Sigma_\lambda^{-1/2-r} \Sigma^{1+2r} \Sigma_\lambda^{-1/2-r}\| \leqslant 1. \qquad (14)$$

**Additional decompositions.** We further bound $\|\mathcal{L}^{-s} Sr_\lambda(\hat{\Sigma}_n)\|$ and $\|\mathcal{L}^{-s} Sq_\lambda(\hat{\Sigma}_n)\Sigma_\lambda^{1/2}\|$. For the first, by the identity $\mathcal{L}^{-s} Sr_\lambda(\hat{\Sigma}_n) = \mathcal{L}^{-s} S\widehat{\Sigma}_{n\lambda}^{-1/2} \widehat{\Sigma}_{n\lambda}^{1/2} r_\lambda(\hat{\Sigma}_n)$, we have

$$\|\mathcal{L}^{-s} Sr_\lambda(\hat{\Sigma}_n)\| = \|\mathcal{L}^{-s} S\widehat{\Sigma}_{n\lambda}^{-1/2}\| \|\widehat{\Sigma}_{n\lambda}^{1/2} r_\lambda(\hat{\Sigma}_n)\|,$$

where

$$\|\widehat{\Sigma}_{n\lambda}^{1/2} r_\lambda(\hat{\Sigma}_n)\| = \sup_{\sigma \in \sigma(\hat{\Sigma}_n)} (\sigma + \lambda)^{1/2} r_\lambda(\sigma) \leqslant \sup_{\sigma \geqslant 0}(\sigma + \lambda)^{1/2} r_\lambda(\sigma) \leqslant 2c_q \lambda^{1/2}.$$

Similarly, by using the identity

$$\mathcal{L}^{-s} Sq_\lambda(\hat{\Sigma}_n)\Sigma_\lambda^{1/2} = \mathcal{L}^{-s} S\widehat{\Sigma}_{n\lambda}^{-1/2} \, \widehat{\Sigma}_{n\lambda}^{1/2} q_\lambda(\hat{\Sigma}_n)\widehat{\Sigma}_{n\lambda}^{1/2} \, \widehat{\Sigma}_{n\lambda}^{-1/2}\Sigma_\lambda^{1/2},$$

we have

$$\|\mathcal{L}^{-s} Sq_\lambda(\hat{\Sigma}_n)\Sigma_\lambda^{1/2}\| = \|\mathcal{L}^{-s} S\widehat{\Sigma}_{n\lambda}^{-1/2}\| \, \|\widehat{\Sigma}_{n\lambda}^{1/2} q_\lambda(\hat{\Sigma}_n)\widehat{\Sigma}_{n\lambda}^{1/2}\| \, \|\widehat{\Sigma}_{n\lambda}^{-1/2}\Sigma_\lambda^{1/2}\|.$$

Finally note that

$$\|\mathcal{L}^{-s} S\widehat{\Sigma}_{n\lambda}^{-1/2}\| \leqslant \|\mathcal{L}^{-s} S\Sigma_\lambda^{-1/2+s}\| \|\Sigma_\lambda^{-s}\| \|\Sigma_\lambda^{1/2}\widehat{\Sigma}_{n\lambda}^{-1/2}\|,$$

and $\|\mathcal{L}^{-s} S\Sigma_\lambda^{-1/2+s}\| \leqslant 1$, $\|\Sigma_\lambda^{-s}\| \leqslant \lambda^{-s}$, and moreover

$$\|\widehat{\Sigma}_{n\lambda}^{1/2} q_\lambda(\hat{\Sigma}_n)\widehat{\Sigma}_{n\lambda}^{1/2}\| = \sup_{\sigma \in \sigma(\hat{\Sigma}_n)} (\sigma + \lambda)q_\lambda(\sigma) \leqslant \sup_{\sigma \geqslant 0}(\sigma + \lambda)q_\lambda(\sigma) \leqslant 2c_q,$$

so, in conclusion

$$\|\mathcal{L}^{-s} Sr_\lambda(\hat{\Sigma}_n)\| \leqslant 2c_q \lambda^{1/2-s}\beta, \quad \|\mathcal{L}^{-s} Sq_\lambda(\hat{\Sigma}_n)\Sigma_\lambda^{1/2}\| \leqslant 2c_q \lambda^{-s}\beta^2.$$

The final result is obtained by gathering the upper bounds for the three terms above and the additional terms of this last section. $\qquad \square$

**Lemma 11.** *Let $\lambda > 0$ and $s \in (0, \min(r, 1/2)]$. Under Assumption (A5) (see Rem. 3), the following holds*

$$\|\mathcal{L}^{-s}(S\widehat{g}_\lambda - Pf_\rho)\|_{L_2(d\rho_{\mathcal{X}})} \leqslant \lambda^{r-s}\|\phi\|_{L_2(d\rho_{\mathcal{X}})}.$$

*Proof.* Since $S\Sigma_\lambda^{-1}S^* = \mathcal{L}\mathcal{L}_\lambda^{-1} = I - \lambda\mathcal{L}_\lambda^{-1}$, we have

$$\begin{aligned}
\mathcal{L}^{-s}(Sg_\lambda - Pf_\rho) &= \mathcal{L}^{-s}(S\Sigma_\lambda^{-1}S^*f_\rho - Pf_\rho) = \mathcal{L}^{-s}(S\Sigma_\lambda^{-1}S^*Pf_\rho - Pf_\rho)\\
&= \mathcal{L}^{-s}(S\Sigma_\lambda^{-1}S^* - I)Pf_\rho = \mathcal{L}^{-s}(S\Sigma_\lambda^{-1}S^* - I)\mathcal{L}^r\phi\\
&= -\lambda\mathcal{L}^{-s}\mathcal{L}_\lambda^{-1}L^r\phi = -\lambda^{r-s}\,\lambda^{1-r+s}\mathcal{L}_\lambda^{-(1-r+s)}\,\mathcal{L}_\lambda^{-(r-s)}\mathcal{L}^{r-s}\,\phi,
\end{aligned}$$

from which

$$\begin{aligned}
\|\mathcal{L}^{-s}(Sg_\lambda - Pf_\rho)\|_{L_2(d\rho_{\mathcal{X}})} &\leqslant \lambda^{r-s}\|\lambda^{1-r+s}\mathcal{L}_\lambda^{-(1-r+s)}\|\|\mathcal{L}_\lambda^{-(r-s)}\mathcal{L}^{r-s}\|\,\|\phi\|_{L_2(d\rho_{\mathcal{X}})}\\
&\leqslant \lambda^{r-s}\|\phi\|_{L_2(d\rho_{\mathcal{X}})}.
\end{aligned}$$

$\square$

**Theorem 2.** *Let $\lambda > 0$ and $s \in (0, \min(r, 1/2)]$. Under Assumption (A5) (see Rem. 3), the following holds*

$$\|\mathcal{L}^{-s}(S\widehat{g}_\lambda - Pf_\rho)\|_{L_2(d\rho_{\mathcal{X}})} \leqslant 2\lambda^{-s}\beta^2 c_q\|\Sigma_\lambda^{-1/2}(\hat{S}_n^*\hat{y} - \hat{\Sigma}_n g_\lambda)\|_{\mathcal{H}} + \left(1 + \beta 2c_q\|\phi\|_{L_2(d\rho_{\mathcal{X}})}\right)\lambda^{r-s}$$

*where $\beta := \|\Sigma_\lambda^{1/2}\widehat{\Sigma}_{n\lambda}^{-1/2}\|$.*

*Proof.* By Prop. 2, we can characterize the excess risk of $\widehat{g}_\lambda$ in terms of the $L_2(d\rho_{\mathcal{X}})$ squared norm of $S\widehat{g}_\lambda - Pf_\rho$. In this paper, simplifying the analysis of [14], we perform the following decomposition

$$\begin{aligned}
\mathcal{L}^{-s}(S\widehat{g}_\lambda - Pf_\rho) &= \mathcal{L}^{-s}S\widehat{g}_\lambda - \mathcal{L}^{-s}Sg_\lambda\\
&\quad + \mathcal{L}^{-s}(Sg_\lambda - Pf_\rho).
\end{aligned}$$

The first term is bounded by Lemma 10, the second is bounded by Lemma 11. $\square$

### D.3 Probabilistic bounds

In this section denote by $\mathcal{N}_\infty(\lambda)$, the quantity

$$\mathcal{N}_\infty(\lambda) = \sup_{x \in S}\|\Sigma_\lambda^{-1/2}K_x\|_{\mathcal{H}}^2,$$

where $S \subseteq \mathcal{X}$ is the support of the probability measure $\rho_{\mathcal{X}}$.

**Lemma 12.** *Under Asm. (A3), we have that for any $g \in \mathcal{H}$*

$$\sup_{x \in supp(\rho_{\mathcal{X}})}|g(x)| \leqslant \kappa_\mu R^\mu\|\Sigma^{1/2(1-\mu)}g\|_{\mathcal{H}} = \kappa_\mu R^\mu\|\mathcal{L}^{-\mu/2}Sg\|_{L_2(d\rho_{\mathcal{X}})}.$$

*Proof.* Note that, Asm. (A3) is equivalent to

$$\|\Sigma^{-1/2(1-\mu)}K_x\| \leqslant \kappa_\mu R^\mu,$$

for all $x$ in the support of $\rho_{\mathcal{X}}$. Then we have, for any $x$ in the support of $\rho_{\mathcal{X}}$,

$$\begin{aligned}
|g(x)| = \langle g, K_x\rangle_{\mathcal{H}} &= \left\langle \Sigma^{1/2(1-\mu)}g, \Sigma^{-1/2(1-\mu)}K_x\right\rangle_{\mathcal{H}}\\
&\leqslant \|\Sigma^{1/2(1-\mu)}g\|_{\mathcal{H}}\|\Sigma^{-1/2(1-\mu)}K_x\| \leqslant \kappa_\mu R^\mu\|\Sigma^{1/2(1-\mu)}g\|_{\mathcal{H}}.
\end{aligned}$$

Now note that, since $\Sigma^{1-\mu} = S^*\mathcal{L}^{-\mu}S$, we have

$$\|\Sigma^{1/2(1-\mu)}g\|_{\mathcal{H}}^2 = \langle g, \Sigma^{1-\mu}g\rangle_{\mathcal{H}} = \left\langle \mathcal{L}^{-\mu/2}Sg, \mathcal{L}^{-\mu/2}Sg\right\rangle_{L_2(d\rho_{\mathcal{X}})}.$$

$\square$

**Lemma 13.** *Under Assumption (A3), we have*

$$\mathcal{N}_\infty(\lambda) \leqslant \kappa_\mu^2 R^{2\mu}\lambda^{-\mu}.$$

*Proof.* First denote with $f_{\lambda,u} \in \mathcal{H}$ the function $\Sigma_\lambda^{-1/2} u$ for any $u \in \mathcal{H}$ and $\lambda > 0$. Note that

$$\|f_{\lambda,u}\|_{\mathcal{H}} = \|\Sigma_\lambda^{-1/2} u\|_{\mathcal{H}} \leqslant \|\Sigma_\lambda^{-1/2}\| \|u\|_{\mathcal{H}} \leqslant \lambda^{-1/2} \|u\|_{\mathcal{H}}.$$

Moreover, since for any $g \in \mathcal{H}$ the identity $\|g\|_{L_2(d\rho_{\mathcal{X}})} = \|Sg\|_{\mathcal{H}}$, we have

$$\|f_{\lambda,u}\|_{L_2(d\rho_{\mathcal{X}})} = \|S\Sigma_\lambda^{-1/2} u\|_{\mathcal{H}} \leqslant \|S\Sigma_\lambda^{-1/2}\| \|u\|_{\mathcal{H}} \leqslant \|u\|_{\mathcal{H}}.$$

Now denote with $B(\mathcal{H})$ the unit ball in $\mathcal{H}$, by applying Asm. (A3) to $f_{\lambda,u}$ we have that

$$\mathcal{N}_\infty(\lambda) = \sup_{x \in S} \|\Sigma_\lambda^{-1/2} K_x\|^2 = \sup_{x \in S, u \in B(\mathcal{H})} \left\langle u, \Sigma_\lambda^{-1/2} K_x \right\rangle_{\mathcal{H}}^2$$

$$= \sup_{x \in S, u \in B(\mathcal{H})} \langle f_{\lambda,u}, K_x \rangle_{\mathcal{H}}^2 = \sup_{u \in B(\mathcal{H})} \sup_{x \in S} |f_{\lambda,u}(x)|^2$$

$$\leqslant \kappa_\mu^2 R^{2\mu} \sup_{u \in B(\mathcal{H})} \|f_{\lambda,u}\|_{\mathcal{H}}^{2\mu} \|f_{\lambda,u}\|_{L_2(d\rho_{\mathcal{X}})}^{2-2\mu}$$

$$\leqslant \kappa_\mu^2 R^{2\mu} \lambda^{-\mu} \sup_{u \in B(\mathcal{H})} \|u\|_{\mathcal{H}}^2 \leqslant \kappa_\mu^2 R^{2\mu} \lambda^{-\mu}.$$

$\square$

**Lemma 14.** *Let $\lambda > 0$, $\delta \in (0,1]$ and $n \in \mathbb{N}$. Under Assumption (A3), we have that, when*

$$n \geqslant 11(1 + \kappa_\mu^2 R^{2\mu} \lambda^{-\mu}) \log \frac{8R^2}{\lambda\delta},$$

*then the following holds with probability $1 - \delta$,*

$$\|\Sigma_\lambda^{1/2} \widehat{\Sigma}_{n\lambda}^{-1/2}\|^2 \leqslant 2.$$

*Proof.* This result is a refinement of the one in [39] and is based on non-commutative Bernstein inequalities for random matrices [40]. By Prop. 8 in [21], we have that

$$\|\Sigma_\lambda^{1/2} \widehat{\Sigma}_{n\lambda}^{-1/2}\|^2 \leqslant (1-t)^{-1}, \quad t := \|\Sigma_\lambda^{-1/2}(\Sigma - \hat{\Sigma}_n)\Sigma_\lambda^{-1/2}\|.$$

When $0 < \lambda \leqslant \|\Sigma\|$, by Prop. 6 of [21] (see also [41] Lemma 9 for more refined constants), we have that the following holds with probability at least $1 - \delta$,

$$t \leqslant \frac{2\eta(1 + \mathcal{N}_\infty(\lambda))}{3n} + \sqrt{\frac{2\eta\mathcal{N}_\infty(\lambda)}{n}},$$

with $\eta = \log \frac{8R^2}{\lambda\delta}$. Finally, by selecting $n \geqslant 11(1 + \kappa_\mu^2 R^{2\mu} \lambda^{-\mu})\eta$, we have that $t \leqslant 1/2$ and so $\|\Sigma_\lambda^{1/2} \widehat{\Sigma}_{n\lambda}^{-1/2}\|^2 \leqslant (1-t)^{-1} \leqslant 2$, with probability $1 - \delta$.

To conclude note that when $\lambda \geqslant \|\Sigma\|$, we have

$$\|\Sigma_\lambda^{1/2} \widehat{\Sigma}_{n\lambda}^{-1/2}\|^2 \leqslant \|\Sigma + \lambda I\| \|(\hat{\Sigma}_n + \lambda I)^{-1}\| \leqslant \frac{\|\Sigma\| + \lambda}{\lambda} = 1 + \frac{\|\Sigma\|}{\lambda} \leqslant 2.$$

$\square$

**Lemma 15.** *Under Assumption (A3), (A4), (A5) (see Rem. 3), (A6) we have*

1. *Let $\lambda > 0$, $n \in \mathbb{N}$, the following holds*

$$\mathbb{E}[\|\Sigma_\lambda^{-1/2}(\hat{S}_n^* \hat{y} - \hat{\Sigma}_n g_\lambda)\|_{\mathcal{H}}^2] \leqslant \|\phi\|_{L_2(d\rho_{\mathcal{X}})}^2 \lambda^{2r} + \frac{2\kappa_\mu^2 R^{2\mu} \lambda^{-(\mu-2r)}}{n} + \frac{4\kappa_\mu^2 R^{2\mu} A Q \lambda^{-\frac{q+\mu\alpha}{q\alpha+\alpha}}}{n},$$

*where $A := \|f_\rho - Pf_\rho\|_{L^{2q}(\mathcal{X},\rho_{\mathcal{X}})}^{2-2/(q+1)}$.*

2. *Let $\delta \in (0,1]$, under the same assumptions, the following holds with probability at least $1 - \delta$*

$$\|\Sigma_\lambda^{-1/2}(\hat{S}_n^*\hat{y} - \hat{\Sigma}_n g_\lambda)\|_{\mathcal{H}} \leqslant c_0 \lambda^r + \frac{4(c_1 \lambda^{-\frac{\mu}{2}} + c_2 \lambda^{-r-\mu}) \log \frac{2}{\delta}}{n}$$

$$+ \sqrt{\frac{16 \kappa_\mu^2 R^{2\mu}(\lambda^{-(\mu-2r)} + 2AQ\lambda^{-\frac{q+\mu\alpha}{q\alpha+\alpha}}) \log \frac{2}{\delta}}{n}},$$

*with $c_0 = \|\phi\|_{L_2(d\rho_{\mathcal{X}})}$, $c_1 = \kappa_\mu R^\mu M + \kappa_\mu^2 R^{2\mu}(2R)^{2r-\mu}\|\phi\|_{L_2(d\rho_{\mathcal{X}})}$, $c_2 = \kappa_\mu^2 R^{2\mu}\|\phi\|_{L_2(d\rho_{\mathcal{X}})}$*

*Proof.* First denote with $\zeta_i$ the random variable

$$\zeta_i = (y_i - g_\lambda(x_i))\Sigma_\lambda^{-1/2}K_{x_i}.$$

In particular note that, by using the definitions of $\hat{S}_n$, $\hat{y}$ and $\hat{\Sigma}_n$, we have

$$\Sigma_\lambda^{-1/2}(\hat{S}_n^*\hat{y} - \hat{\Sigma}_n g_\lambda) = \Sigma_\lambda^{-1/2}(\frac{1}{n}\sum_{i=1}^n K_{x_i}y_i - \frac{1}{n}(K_{x_i} \otimes K_{x_i})g_\lambda) = \frac{1}{n}\sum_{i=1}^n \zeta_i.$$

I So, by noting that $\zeta_i$ are independent and identically distributed, we have

$$\mathbb{E}[\|\Sigma_\lambda^{-1/2}(\hat{S}_n^*\hat{y} - \hat{\Sigma}_n g_\lambda)\|_{\mathcal{H}}^2] = \mathbb{E}[\|\frac{1}{n}\sum_{i=1}^n \zeta_i\|_{\mathcal{H}}^2] = \frac{1}{n^2}\sum_{i,j=1}^n \mathbb{E}[\langle \zeta_i, \zeta_j \rangle_{\mathcal{H}}]$$

$$= \frac{1}{n}\mathbb{E}[\|\zeta_1\|_{\mathcal{H}}^2] + \frac{n-1}{n}\|\mathbb{E}[\zeta_1]\|_{\mathcal{H}}^2.$$

Now note that

$$\mathbb{E}[\zeta_1] = \Sigma_\lambda^{-1/2}(\mathbb{E}[K_{x_1}y_1] - \mathbb{E}[K_{x_1} \otimes K_{x_1}]g_\lambda) = \Sigma_\lambda^{-1/2}(S^*f_\rho - \Sigma g_\lambda).$$

In particular, by the fact that $S^*f_\rho = Pf_\rho$, $Pf_\rho = \mathcal{L}^r\phi$ and $\Sigma g_\lambda = \Sigma\Sigma_\lambda^{-1}S^*f_\rho$ and $\Sigma\Sigma_\lambda^{-1} = I - \lambda\Sigma_\lambda^{-1}$, we have

$$\Sigma_\lambda^{-1/2}(S^*f_\rho - \Sigma g_\lambda) = \lambda\Sigma_\lambda^{-3/2}S^*f_\rho = \lambda^r \lambda^{1-r}\Sigma_\lambda^{-(1-r)} \Sigma_\lambda^{-1/2-r}S^*\mathcal{L}^r \phi.$$

So, since $\|\Sigma_\lambda^{-1/2-r}S^*\mathcal{L}^r\| \leqslant 1$, as proven in Eq. 14, then

$$\|\mathbb{E}[\zeta_1]\|_{\mathcal{H}} \leqslant \lambda^r\|\lambda^{1-r}\Sigma_\lambda^{-(1-r)}\| \|\Sigma_\lambda^{-1/2-r}S^*\mathcal{L}^r\| \|\phi\|_{L_2(d\rho_{\mathcal{X}})} \leqslant \lambda^r\|\phi\|_{L_2(d\rho_{\mathcal{X}})} := Z.$$

Morever

$$\mathbb{E}[\|\zeta_1\|_{\mathcal{H}}^2] = \mathbb{E}[\|\Sigma_\lambda^{-1/2}K_{x_1}\|_{\mathcal{H}}^2(y_1 - g_\lambda(x_1))^2] = \mathbb{E}_{x_1}\mathbb{E}_{y_1|x_1}[\|\Sigma_\lambda^{-1/2}K_{x_1}\|_{\mathcal{H}}^2(y_1 - g_\lambda(x_1))^2]$$

$$= \mathbb{E}_{x_1}[\|\Sigma_\lambda^{-1/2}K_{x_1}\|_{\mathcal{H}}^2(f_\rho(x_1) - g_\lambda(x_1))^2].$$

Moreover we have

$$\mathbb{E}[\|\zeta_1\|_{\mathcal{H}}^2] = \mathbb{E}_x[\|\Sigma_\lambda^{-1/2}K_x\|_{\mathcal{H}}^2(f_\rho(x) - g_\lambda(x))^2]$$

$$= \mathbb{E}_x[\|\Sigma_\lambda^{-1/2}K_x\|_{\mathcal{H}}^2((f_\rho(x) - (Pf_\rho)(x)) + ((Pf_\rho)(x) - g_\lambda(x)))^2]$$

$$\leqslant 2\mathbb{E}_x[\|\Sigma_\lambda^{-1/2}K_x\|_{\mathcal{H}}^2(f_\rho(x) - (Pf_\rho)(x))^2] + 2\mathbb{E}_x[\|\Sigma_\lambda^{-1/2}K_x\|_{\mathcal{H}}^2((Pf_\rho)(x) - g_\lambda(x))^2].$$

Now since $\mathbb{E}[AB] \leqslant (\text{ess sup } A)\mathbb{E}[B]$, for any two random variables $A, B$, we have

$$\mathbb{E}_x[\|\Sigma_\lambda^{-1/2}K_x\|_{\mathcal{H}}^2((Pf_\rho)(x) - g_\lambda(x))^2] \leqslant \mathcal{N}_\infty(\lambda)\mathbb{E}_x[((Pf_\rho)(x) - g_\lambda(x))^2]$$

$$= \mathcal{N}_\infty(\lambda)\|Pf_\rho - Sg_\lambda\|_{L_2(d\rho_{\mathcal{X}})}^2$$

$$\leqslant \kappa_\mu^2 R^{2\mu}\lambda^{-(\mu-2r)},$$

where in the last step we bounded $\mathcal{N}_\infty(\lambda)$ via Lemma 13 and $\|Pf_\rho - Sg_\lambda\|_{L_2(d\rho_{\mathcal{X}})}^2$, via Lemma. 11 applied with $s = 0$. Finally, denoting by $a(x) = \|\Sigma_\lambda^{-1/2}K_x\|_{\mathcal{H}}^2$ and $b(x) = (f_\rho(x) - (Pf_\rho)(x))^2$

and noting that by Markov inequality we have $\mathbb{E}_x[\mathbf{1}_{\{b(x)>t\}}] = \rho_{\mathcal{X}}(\{b(x) > t\}) = \rho_{\mathcal{X}}(\{b(x)^q > t^q\}) \leqslant \mathbb{E}_x[b(x)^q]t^{-q}$, for any $t > 0$. Then for any $t > 0$ the following holds

$$\begin{aligned} \mathbb{E}_x[a(x)b(x)] &= \mathbb{E}_x[a(x)b(x)\mathbf{1}_{\{b(x)\leqslant t\}}] + \mathbb{E}_x[a(x)b(x)\mathbf{1}_{\{b(x)>t\}}] \\ &\leqslant t\mathbb{E}_x[a(x)] + N_\infty(\lambda)\mathbb{E}_x[b(x)\mathbf{1}_{\{b(x)>t\}}] \\ &\leqslant tN(\lambda) + N_\infty(\lambda)\mathbb{E}_x[b(x)^q]t^{-q}. \end{aligned}$$

By minimizing the quantity above in $t$, we obtain

$$\mathbb{E}_x[\|\Sigma_\lambda^{-1/2}K_x\|_{\mathcal{H}}^2(f_\rho(x) - (Pf_\rho)(x))^2] \leqslant 2\|f_\rho - Pf_\rho\|_{L^q(X,\rho_{\mathcal{X}})}^{\frac{q}{q+1}}\mathcal{N}(\lambda)^{\frac{q}{q+1}}\mathcal{N}_\infty(\lambda)^{\frac{1}{q+1}}$$

$$\leqslant 2\kappa_\mu^2 R^{2\mu}AQ\lambda^{-\frac{q+\mu\alpha}{q\alpha+\alpha}}.$$

So finally

$$\mathbb{E}[\|\zeta_1\|_{\mathcal{H}}^2] \leqslant 2\kappa_\mu^2 R^{2\mu}\lambda^{-(\mu-2r)} + 4\kappa_\mu^2 R^{2\mu}AQ\lambda^{-\frac{q+\mu\alpha}{q\alpha+\alpha}} := W^2.$$

To conclude the proof, let us obtain the bound in high probability. We need to bound the higher moments of $\zeta_1$. First note that

$$\mathbb{E}[\|\zeta_1 - \mathbb{E}[\zeta_1]\|_{\mathcal{H}}^p] \leqslant \mathbb{E}[\|\zeta_1 - \zeta_2\|_{\mathcal{H}}^p] \leqslant 2^{p-1}\mathbb{E}[\|\zeta_1\|_{\mathcal{H}}^p + \|\zeta_2\|_{\mathcal{H}}^p] \leqslant 2^p\mathbb{E}[\|\zeta_1\|_{\mathcal{H}}^p].$$

Moreover, denoting by $S \subseteq \mathcal{X}$ the support of $\rho_{\mathcal{X}}$ and recalling that $y$ is bounded in $[-M, M]$, the following bound holds almost surely

$$\begin{aligned} \|\zeta_1\| &\leqslant \sup_{x\in S}\|\Sigma_\lambda^{-1/2}K_x\|(M + |g_\lambda(x)|) \leqslant (\sup_{x\in S}\|\Sigma_\lambda^{-1/2}K_x\|)(M + \sup_{x\in S}|g_\lambda(x)|) \\ &\leqslant \kappa_\mu R^\mu\lambda^{-\mu/2}(M + \kappa_\mu R^\mu\|\Sigma^{1/2(1-\mu)}g_\lambda\|_{\mathcal{H}}). \end{aligned}$$

where in the last step we applied Lemma 13 and Lemma 12. In particular, by definition of $g_\lambda$, the fact that $S^*f_\rho = S^*Pf_\rho$, that $Pf_\rho = \mathcal{L}^r\phi$ and that $\|\Sigma_\lambda^{-(1/2+r)}S^*\mathcal{L}^r\| \leqslant 1$ as proven in Eq. 14, we have

$$\begin{aligned} \|\Sigma^{1/2(1-\mu)}g_\lambda\|_{\mathcal{H}} &= \|\Sigma^{1/2(1-\mu)}\Sigma_\lambda^{-1}S^*\mathcal{L}^r\phi\|_{\mathcal{H}} \\ &\leqslant \|\Sigma^{1/2(1-\mu)}\Sigma^{-1/2(1-\mu)}\|\|\Sigma_\lambda^{-(\mu/2-r)}\|\|\Sigma_\lambda^{-(1/2+r)}S^*\mathcal{L}^r\|\|\phi\|_{L_2(d\rho_{\mathcal{X}})} \\ &\leqslant \|\Sigma_\lambda^{r-\mu/2}\|\|\phi\|_{L_2(d\rho_{\mathcal{X}})}. \end{aligned}$$

Finally note that if $r \leqslant \mu/2$ then $\|\Sigma_\lambda^{r-\mu/2}\| \leqslant \lambda^{-(\mu/2-r)}$, if $r \geqslant \mu/2$ then

$$\|\Sigma_\lambda^{r-\mu/2}\| = (\|C\| + \lambda)^{r-\mu/2} \leqslant (2\|C\|)^{r-\mu/2} \leqslant (2R)^{2r-\mu}.$$

So in particular

$$\|\Sigma_\lambda^{r-\mu/2}\| \leqslant (2R)^{2r-\mu} + \lambda^{-(\mu/2-r)}.$$

Then the following holds almost surely

$$\|\zeta_1\| \leqslant (\kappa_\mu R^\mu M + \kappa_\mu^2 R^{2\mu}(2R)^{2r-\mu}\|\phi\|_{L_2(d\rho_{\mathcal{X}})})\lambda^{-\mu/2} + \kappa_\mu^2 R^{2\mu}\|\phi\|_{L_2(d\rho_{\mathcal{X}})}\lambda^{r-\mu} := V.$$

So finally

$$\mathbb{E}[\|\zeta_1 - \mathbb{E}[\zeta_1]\|_{\mathcal{H}}^p] \leqslant 2^p\mathbb{E}[\|\zeta_1\|_{\mathcal{H}}^p] \leqslant \frac{p!}{2}(2V)^{p-2}(4W^2).$$

By applying Pinelis inequality, the following holds with probability $1 - \delta$

$$\|\frac{1}{n}\sum_{i=1}^n(\zeta_i - \mathbb{E}[\zeta_i])\|_{\mathcal{H}} \leqslant \frac{4V\log\frac{2}{\delta}}{n} + \sqrt{\frac{8W\log\frac{2}{\delta}}{n}}.$$

So with the same probability

$$\|\frac{1}{n}\sum_{i=1}^n\zeta_i\|_{\mathcal{H}} \leqslant \|\frac{1}{n}\sum_{i=1}^n(\zeta_i - \mathbb{E}[\zeta_i])\|_{\mathcal{H}} + \|\mathbb{E}[\zeta_1]\|_{\mathcal{H}} \leqslant Z + \frac{4V\log\frac{2}{\delta}}{n} + \sqrt{\frac{8W\log\frac{2}{\delta}}{n}}.$$

$\square$

**Lemma 16.** *Let $\lambda > 0$, $n \in \mathbb{N}$ and $s \in (0, 1/2]$. Let $\delta \in (0, 1]$. Under Assumption (A3), (A4), (A5) (see Rem. 3), (A6), when*

$$n \geqslant 11(1 + \kappa_\mu^2 R^{2\mu} \lambda^{-\mu}) \log \frac{16R^2}{\lambda\delta},$$

*then the following holds with probability $1 - \delta$,*

$$\|\mathcal{L}^{-s} S(\widehat{g}_\lambda - g_\lambda)\|_{L_2(d\rho_{\mathcal{X}})} \leq c_0 \lambda^{r-s} + \frac{(c_1 \lambda^{-\frac{\mu}{2}-s} + c_2 \lambda^{r-\mu-s}) \log \frac{4}{\delta}}{n}$$
$$+ \sqrt{\frac{(c_3 \lambda^{-(\mu+2s-2r)} + c_4 \lambda^{-\frac{q+\mu\alpha}{q\alpha+\alpha}-2s}) \log \frac{4}{\delta}}{n}}.$$

*with $c_0 = 7c_q \|\phi\|_{L_2(d\rho_{\mathcal{X}})}$, $c_1 = 16c_q(\kappa_\mu R^\mu M + \kappa_\mu^2 R^{2\mu}(2R)^{2r-\mu}\|\phi\|_{L_2(d\rho_{\mathcal{X}})})$, $c_2 = 16c_q\kappa_\mu^2 R^{2\mu}\|\phi\|_{L_2(d\rho_{\mathcal{X}})}$, $c_3 = 64\kappa_\mu^2 R^{2\mu} c_q^2$, $c_4 = 128\kappa_\mu^2 R^{2\mu} AQ c_q^2$.*

*Proof.* Let $\tau = \delta/2$, the result is obtained by combining Lemma 10, with Lemma 15 with probability $\tau$, and Lemma 14, with probability $\tau$ and then taking the intersection bound of the two events. $\square$

**Corollary 1.** *Let $\lambda > 0$, $n \in \mathbb{N}$ and $s \in (0, 1/2]$. Let $\delta \in (0, 1]$. Under the assumptions of Lemma 16, when*

$$n \geqslant 11(1 + \kappa_\mu^2 R^{2\mu} \lambda^{-\mu}) \log \frac{16R^2}{\lambda\delta},$$

*then the following holds with probability $1 - \delta$,*

$$\|\mathcal{L}^{-s} S\widehat{g}_\lambda\|_{L_2(d\rho_{\mathcal{X}})} \leq R^{2r-2s} + (1 + c_0)\lambda^{r-s} + \frac{(c_1 \lambda^{-\frac{\mu}{2}-s} + c_2 \lambda^{r-\mu-s}) \log \frac{4}{\delta}}{n}$$
$$+ \sqrt{\frac{(c_3 \lambda^{-(\mu+2s-2r)} + c_4 \lambda^{-\frac{q+\mu\alpha}{q\alpha+\alpha}-2s}) \log \frac{4}{\delta}}{n}} +$$

*with the same constants $c_0, \dots, c_4$ as in Lemma 16.*

*Proof.* First note that

$$\|\mathcal{L}^{-s} S\widehat{g}_\lambda\|_{L_2(d\rho_{\mathcal{X}})} \leqslant \|\mathcal{L}^{-s} S(\widehat{g}_\lambda - g_\lambda)\|_{L_2(d\rho_{\mathcal{X}})} + \|\mathcal{L}^{-s} Sg_\lambda\|_{L_2(d\rho_{\mathcal{X}})}.$$

The first term on the right hand side is controlled by Lemma 16, for the second, by using the definition of $g_\lambda$ and Asm. (A5) (see Rem. 3), we have

$$\|\mathcal{L}^{-s} Sg_\lambda\|_{L_2(d\rho_{\mathcal{X}})} \leqslant \|\mathcal{L}^{-s} S\Sigma_\lambda^{-1/2+s}\| \|\Sigma_\lambda^{-(s-r)}\| \|\Sigma_\lambda^{-1/2-r} S^* \mathcal{L}^r\| \|\phi\|_{L_2(d\rho_{\mathcal{X}})}$$
$$\leqslant \|\Sigma_\lambda^{r-s}\| \|\phi\|_{L_2(d\rho_{\mathcal{X}})},$$

where $\|\Sigma_\lambda^{-1/2-r} S^* \mathcal{L}^r\| \leqslant 1$ by Eq. 14 and analogously $\|\mathcal{L}^{-s} S\Sigma_\lambda^{-1/2+s}\| \leqslant 1$. Note that if $s \geqslant r$ then $\|\Sigma_\lambda^{r-s}\| \leqslant \lambda^{-(s-r)}$. If $s < r$, we have

$$\|\Sigma_\lambda^{r-s}\| = (\|\Sigma\| + \lambda)^{r-s} \leqslant \|C\|^{r-s} + \lambda^{r-s} \leqslant R^{2r-2s} + \lambda^{r-s}.$$

So finally $\|\Sigma_\lambda^{r-s}\| \leqslant R^{2r-2s} + \lambda^{r-s}$. $\square$

**Corollary 2.** *Let $\lambda > 0$, $n \in \mathbb{N}$ and $s \in (0, 1/2]$. Let $\delta \in (0, 1]$. Under Assumption (A3), (A4), (A5) (see Rem. 3), (A6), when*

$$n \geqslant 11(1 + \kappa_\mu^2 R^{2\mu} \lambda^{-\mu}) \log \frac{16R^2}{\lambda\delta},$$

*then the following holds with probability $1 - \delta$,*

$$\sup_{x \in \mathcal{X}} |\widehat{g}_\lambda(x)| \leq \kappa_\mu R^\mu R^{2r-2s} + \kappa_\mu R^\mu (1 + c_0)\lambda^{r-\mu/2} + \kappa_\mu R^\mu \frac{(c_1 \lambda^{-\mu} + c_2 \lambda^{r-3/2\mu}) \log \frac{4}{\delta}}{n}$$
$$+ \kappa_\mu R^\mu \sqrt{\frac{(c_3 \lambda^{-(2\mu-2r)} + \kappa_\mu R^\mu c_4 \lambda^{-\frac{q+\mu\alpha}{q\alpha+\alpha}-\mu}) \log \frac{4}{\delta}}{n}}.$$

*with the same constants $c_0, \dots, c_4$ in Lemma 16.*

*Proof.* The proof is obtained by applying Lemma 12 on $\widehat{g}_\lambda$ and then Corollary 1. $\square$

## D.4  Main Result

**Theorem 3.** *Let $\lambda > 0$, $n \in \mathbb{N}$ and $s \in (0, \min(r, 1/2)]$. Under Assumption (A3), (A4), (A5) (see Rem. 3), (A6), when*

$$n \geqslant 11(1 + \kappa_\mu^2 R^{2\mu} \lambda^{-\mu}) \log \frac{c_0}{\lambda^{3+4r-4s}},$$

*then*

$$\mathbb{E}[\|\mathcal{L}^{-s}(S\widehat{g}_\lambda - Pf_\rho)\|_{L_2(d\rho_\mathcal{X})}^2] \leqslant c_1 \frac{\lambda^{-(\mu+2s-2r)}}{n} + c_2 \frac{\lambda^{-\frac{q+\mu\alpha}{q\alpha+\alpha}-2s}}{n} + c_3 \lambda^{2r-2s},$$

*where $m_4 = M^4$, $c_0 = 32R^{4-4s}m_4 + 32R^{8-8r-8s}\|\phi\|_{L_2(d\rho_\mathcal{X})}^4$, $c_1 = 16c_q^2\kappa_\mu^2 R^{2\mu}$, $c_2 = 32c_q^2\kappa_\mu^2 R^{2\mu} AQ$, $c_3 = 3 + 8c_q^2\|\phi\|_{L_2(d\rho_\mathcal{X})}^2$.*

*Proof.* Denote by $R(\widehat{g}_\lambda)$, the expected risk $R(\widehat{g}_\lambda) = \mathcal{E}(\widehat{g}_\lambda) - \inf_{g \in \mathcal{H}} \mathcal{E}(g)$. First, note that by Prop. 2, we have

$$R_s(\widehat{g}_\lambda) = \|\mathcal{L}^{-s}(S\widehat{g}_\lambda - Pf_\rho)\|_{L_2(d\rho_\mathcal{X})}^2.$$

Denote by $E$ the event such that $\beta$ as defined in Thm. 2, satisfies $\beta \leqslant 2$. Then we have

$$\mathbb{E}[R_s(\widehat{g}_\lambda)] = \mathbb{E}[R_s(\widehat{g}_\lambda)\mathbf{1}_E] + \mathbb{E}[R(\widehat{g}_\lambda)\mathbf{1}_{E^c}].$$

For the first term, by Thm. 2 and Lemma 15, we have

$$\mathbb{E}[R_s(\widehat{g}_\lambda)\mathbf{1}_E] \leqslant \mathbb{E}[\Big(2\lambda^{-2s}\beta^4 c_q^2 \|\Sigma_\lambda^{-1/2}(\hat{S}_n^* \hat{y} - \hat{\Sigma}_n g_\lambda)\|_\mathcal{H}^2$$
$$+ 2\Big(1 + \beta^2 2c_q^2\|\phi\|_{L_2(d\rho_\mathcal{X})}^2\Big)\lambda^{2r-2s}\Big)\mathbf{1}_E]$$
$$\leqslant 8\lambda^{-2s}c_q^2 \mathbb{E}[\|\Sigma_\lambda^{-1/2}(\hat{S}_n^* \hat{y} - \hat{\Sigma}_n g_\lambda)\|_\mathcal{H}^2] + 2\Big(1 + 4c_q^2\|\phi\|_{L_2(d\rho_\mathcal{X})}^2\Big)\lambda^{2r-2s}$$
$$\leqslant \frac{16c_q^2\kappa_\mu^2 R^{2\mu}\lambda^{-\mu+2r-2s}}{n} + \frac{32c_q^2\kappa_\mu^2 R^{2\mu} AQ\lambda^{-\frac{q+\mu\alpha}{q\alpha+\alpha}-2s}}{n} + \Big(2 + 8c_q^2\|\phi\|_{L_2(d\rho_\mathcal{X})}^2\Big)\lambda^{2r-2s}.$$

For the second term, since $\widehat{\Sigma}_{n\lambda}^{1/2} q_\lambda(\hat{\Sigma}_n)\widehat{\Sigma}_{n\lambda}^{1/2} = \widehat{\Sigma}_{n\lambda} q_\lambda(\hat{\Sigma}_n) \leqslant \sup_{\sigma>0}(\sigma + \lambda)q_\lambda(\sigma) \leqslant c_q$ by definition of filters, and that $Pf_\rho = L^r\phi$, we have

$$R_s(\widehat{g}_\lambda)^{1/2} \leqslant \|\mathcal{L}^{-s}S\widehat{g}_\lambda\|_{L_2(d\rho_\mathcal{X})} + \|\mathcal{L}^{-s}Pf_\rho\|_{L_2(d\rho_\mathcal{X})}$$
$$\leqslant \|\mathcal{L}^{-s}S\|\|\widehat{\Sigma}_{n\lambda}^{-1/2}\|\|\widehat{\Sigma}_{n\lambda}^{1/2}q_\lambda(\hat{\Sigma}_n)\widehat{\Sigma}_{n\lambda}^{1/2}\|\|\widehat{\Sigma}_{n\lambda}^{-1/2}\hat{S}_n^*\|\|\hat{y}\| + \|\mathcal{L}^{-s}\mathcal{L}^r\|\|\phi\|_{L_2(d\rho_\mathcal{X})}$$
$$\leqslant R^{1/2-s}\lambda^{-1/2}\|\hat{y}\| + R^{2r-2s}\|\phi\|_{L_2(d\rho_\mathcal{X})}$$
$$\leqslant \lambda^{-1/2}(R^{1/2-s}(n^{-1}\sum_{i=1}^n y_i) + R^{1+2r-2s}\|\phi\|_{L_2(d\rho_\mathcal{X})}),$$

where the last step is due to the fact that $1 \leqslant \lambda^{-1/2}\|\mathcal{L}\|^{1/2}$ since $\lambda$ satisfies $0 < \lambda \leqslant \|\Sigma\| = \|\mathcal{L}\| \leqslant R^2$. Denote with $\delta$ the quantity $\delta = \lambda^{2+4r-4s}/c_0$. Since $\mathbb{E}[\mathbf{1}_{E^c}]$ corresponds to the probability of the event $E^c$, and, by Lemma 14, we have that $E^c$ holds with probability at most $\delta$ since $n \geqslant 11(1 + \kappa_\mu^2 R^{2\mu}\lambda^{-\mu}) \log \frac{8R^2}{\lambda\delta}$, then we have that

$$\mathbb{E}[R(\widehat{g}_\lambda)\mathbf{1}_{E^c}] \leqslant \mathbb{E}[\|S\widehat{g}_\lambda\|_{L_2(d\rho_\mathcal{X})}^2\mathbf{1}_{E^c}] \leqslant \sqrt{\mathbb{E}[\|S\widehat{g}_\lambda\|_{L_2(d\rho_\mathcal{X})}^4]}\sqrt{\mathbb{E}[\mathbf{1}_{E^c}]}$$
$$\leqslant \sqrt{\frac{4R^{2-4s}n^{-2}(\sum_{i,j=1}^n \mathbb{E}[y_i^2 y_j^2]) + 4R^{4-8r-8s}\|\phi\|_{L_2(d\rho_\mathcal{X})}^4}{\lambda^2}}\sqrt{\delta}$$
$$\leqslant \frac{\sqrt{\delta}}{\lambda}\sqrt{4R^{2-4s}m_4 + 4R^{4-8r-8s}\|\phi\|_{L_2(d\rho_\mathcal{X})}^4}$$
$$= \frac{\sqrt{\delta c_0/(8R^2)}}{\lambda} \leqslant \lambda^{2r-2s}.$$

$\square$

**Corollary 3.** *Let $\lambda > 0$ and $n \in \mathbb{N}$ and $s = 0$. Under Assumption (A3), (A4), (A5) (see Rem. 3), (A6), when*

$$\lambda = B_1 \begin{cases} n^{-\alpha/\left(2r\alpha + 1 + \frac{\mu\alpha - 1}{q+1}\right)} & 2r\alpha + 1 + \frac{\mu\alpha - 1}{q+1} > \mu\alpha \\ n^{-1/\mu}\left(\log B_2 n\right)^{\frac{1}{\mu}} & 2r\alpha + 1 + \frac{\mu\alpha - 1}{q+1} \leqslant \mu\alpha. \end{cases} \tag{15}$$

*then,*

$$\mathbb{E}\,\mathcal{E}(\widehat{g}_\lambda) - \inf_{g \in \mathcal{H}} \mathcal{E}(g) \leqslant B_3 \begin{cases} n^{-2r\alpha/\left(2r\alpha + 1 + \frac{\mu\alpha - 1}{q+1}\right)} & 2r\alpha + 1 + \frac{\mu\alpha - 1}{q+1} > \mu\alpha \\ n^{-2r/\mu} & 2r\alpha + 1 + \frac{\mu\alpha - 1}{q+1} \leqslant \mu\alpha \end{cases} \tag{16}$$

*where $B_2 = 3 \vee (32R^6 m_4)^{\frac{\mu}{3+4r}} B_1^{-\mu}$ and $B_1$ defined explicitly in the proof.*

*Proof.* The proof of this corollary is a direct application of Thm. 3. In the rest of the proof we find the constants to guarantee that the condition relating $n, \lambda$ in the theorem is always satisfied. Indeed to guarantee the applicability of Thm. 3, we need to be sure that $n \geqslant 11(1 + \kappa_\mu^2 R^{2\mu}\lambda^{-\mu}) \log \frac{32R^6 m_4}{\lambda^{3+4r}}$. This is satisfied when both the following conditions hold $n \geqslant 22 \log \frac{32R^6 m_4}{\lambda^{3+4r}}$ and $n \geqslant 2\kappa_\mu^2 R^{2\mu}\lambda^{-\mu} \log \frac{32R^6 m_4}{\lambda^{3+4r}}$. To study the last two conditions, we recall that for $A, B, s, q > 0$ we have that $An^{-s}\log(Bn^q)$ satisfy

$$An^{-s}\log(Bn^q) = \frac{qAB^{s/q}}{s}\frac{\log B^{s/q}n^s}{B^{s/q}n^s} \leqslant \frac{qAB^{s/q}}{es},$$

for any $n > 0$, since $\frac{logx}{x} \leqslant \frac{1}{e}$ for any $x > 0$. Now we define explicitly $B_1$, let $\tau = \alpha/\left(2r\alpha + 1 + \frac{\mu\alpha - 1}{q+1}\right)$, we have

$$B_1 = \left(\frac{22(3+4r)}{e\mu}(32R^6 m_4)^{\frac{\mu}{3+4r}}\right)^{\frac{1}{\mu}} \vee \tag{17}$$

$$\vee \begin{cases} \left(\frac{2M(3+4r)}{e(1/\tau - \mu)}(32R^6 m_4)^{\frac{1/\tau - \mu}{3+4r}}\right)^{\tau} & 2r\alpha + 1 + \frac{\mu\alpha - 1}{q+1} > \mu\alpha \\ \left(\frac{2M(3+4r)}{\mu}\right)^{\frac{1}{\mu}} & 2r\alpha + 1 + \frac{\mu\alpha - 1}{q+1} \leqslant \mu\alpha \end{cases}. \tag{18}$$

For the first condition, we use the fact that $\lambda$ is always larger than $B_1 n^{-1/\mu}$, so we have

$$\frac{22}{n}\log\frac{32R^6 m_4}{\lambda^{3+4r}} \leqslant \frac{22}{n}\log\frac{32R^6 m_4 n^{(3+4r)/\mu}}{B_1^{3+4r}} \leqslant \frac{22(3+4r)(32R^6 m_4)^{\mu/(3+4r)}}{e\mu B_1^\mu} \leqslant 1.$$

For the second inequality, when $2r\alpha + 1 + \frac{\mu\alpha - 1}{q+1} \geqslant \mu\alpha$, we have $\lambda = B_1 n^{-\tau}$, so

$$\frac{2\kappa_\mu^2 R^{2\mu}}{n}\lambda^{-\mu}\log\frac{32R^6 m_4}{\lambda^{3+4r}} \leqslant \frac{2\kappa_\mu^2 R^{2\mu}}{B_1^\mu n^{1-\mu\tau}}\log\frac{32R^6 m_4 n^{(3+4r)\tau}}{B_1^{3+4r}}$$

$$\leqslant \frac{2\kappa_\mu^2 R^{2\mu}(3+4r)\tau}{e(1-\mu\tau)}\frac{(32R^6 m_4)^{\frac{1/\tau - \mu}{3+4r}}}{B_1^{1/\tau}} \leqslant 1.$$

Finally, when $2r\alpha + 1 + \frac{\mu\alpha - 1}{q+1} \geqslant \mu\alpha$, we have $\lambda = B_1 n^{-1/\mu}(\log B_2 n)^{1/\mu}$. So since $\log(B_2 n) > 1$, we have

$$\frac{2\kappa_\mu^2 R^{2\mu}}{n}\log\frac{32R^6 m_4}{\lambda^{3+4r}} \leqslant \frac{2\kappa_\mu^2 R^{2\mu}}{B_1^\mu}\frac{\log\frac{32R^6 m_4 n^{(3+4r)/\mu}}{B_1^{3+4r}}}{\log(B_2 n)} = \frac{2\kappa_\mu^2 R^{2\mu}(3+4r)}{\mu B_1^\mu}\frac{\log\frac{(32R^6 m_4)^{\mu/(3+4r)}n}{B_1^\mu}}{\log(B_2 n)} \leqslant 1.$$

So by selecting $\lambda$ as in Eq. 15, we guarantee that the condition required by Thm. 3 is satisfied.

Finally the constant $B_3$ is obtained by

$$B_3 = c_1 \max(1, w)^{-(\mu + 2s - 2r)} + c_2 \max(1, w)^{-\frac{q+\mu\alpha}{q\alpha + \alpha} - 2s} + c_3 \max(1, w)^{2r - 2s},$$

with $w = B_1 \log(1 + B_2)$ and $c_1, c_2, c_3$ as in Thm. 3. $\qquad\square$

# E Experiments with different sampling

We present here the results for two different types of sampling, which seem to be more stable, perform better and are widely used in practice :

**Without replacement (Figure 4)**: for which we select randomly the data points but never use two times over the same point in one epoch.

**Cycles (Figure 5)**: for which we pick successively the data points in the same order.

Figure 4 − The sampling is performed by **cycling over the data** The four plots represent each a different configuration on the $(\alpha, r)$ plan represented in Figure 1, for $r = 1/(2\alpha)$. **Top left** ($\alpha = 1.5$) and **right** ($\alpha = 2$) are two easy problems (Top right is the limiting case where $r = \frac{\alpha-1}{2\alpha}$) for which one pass over the data is optimal. **Bottom left** ($\alpha = 2.5$) and **right** ($\alpha = 3$) are two hard problems for which an increasing number of passes is recquired. The blue dotted line are the slopes predicted by the theoretical result in Theorem 1.

Figure 5 – The sampling is performed **without replacement**. The four plots represent each a different configuration on the $(\alpha, r)$ plan represented in Figure 1, for $r = 1/(2\alpha)$. **Top left** ($\alpha = 1.5$) and **right** ($\alpha = 2$) are two easy problems (Top right is the limiting case where $r = \frac{\alpha-1}{2\alpha}$) for which one pass over the data is optimal. **Bottom left** ($\alpha = 2.5$) and **right** ($\alpha = 3$) are two hard problems for which an increasing number of passes is recquired. The blue dotted line are the slopes predicted by the theoretical result in Theorem 1.

## Footnotes

[2]Indeed, adapting a similar result from [18], on the one hand, $1 - (1 - \rho)^k \leqslant 1$ implying that $(1 - (1 - \rho)^k)^{1-1/\alpha+u} \leqslant 1$. On the other hand, $1 - (1 - \gamma x)^k \leqslant \gamma k x$ implying that $(1 - (1 - \rho)^k)^{1+1/\alpha-u} \leqslant (k\rho)^{1+1/\alpha-u}$. Thus by multiplying the two we get $(1 - (1 - \rho)^k)^2 \leqslant (k\rho)^{1-u+1/\alpha}$.