[Reviews · NeurIPS 2018]

Reviewer 1



This paper identifies and separates (kernel) linear least-squares regression problems wherein carrying out multiple passes of stochastic gradient descent (SGD) over a training set can yield better statistical error than only a single pass. This is relevant to the core of machine learning theory, and relates to a line of work published at NIPS, ICML, COLT, and similar conferences in the past several years about the statistical error of one-pass, many-pass, and ERM-based learning. The authors focus on regression problems captured, by assumption, by two parameters: alpha, which governs the exponent of a power-law eigenvalue decay, and r, which governs a transformation under which the Hilbert norm of the optimal predictor is bounded. They refer to problems where r <= (alpha - 1) / (2 * alpha) as "hard". The main result of the paper is to show that for these "hard" problems, multiple SGD passes either achieve (minimax) optimal rates of statistical estimation, or at least improve the rate relative to a single pass. The results are interesting and might address an unanswered core question in machine learning, and the mathematical presentation is clear, with assumptions upfront. What's most needed is clarification on whether this applies to core machine learning problems, and how this relates to the previous results on optimality of single-pass SGD. What problem instances are included in the assumptions and then in the different "easy" and "hard" subclasses? In what way were the "hard" problems not also solved by single-pass SGD? Two examples are given in the paper. Example 1 is a learning problem over the one-dimensional unit interval. Example 2 states a family of learning problems for which the authors claim alpha = 2m/d and r = s/(2m), provided d < 2m (indeed if d >= 2m then alpha is nonpositive and so there is no valid r to make for a "hard problem"). These examples left me confused about the introductory claim (e.g. in the abstract) that single pass SGD is better "for low-dimensional easy problems." Both examples are learning problems over input domains of inherently low dimension. Experimentally, least-squares regression is an interesting core problem for the NIPS community. What is a reasonable choice of the parameters alpha and r for different common datasets, or even simply MNIST? What if a feature map or kernel known to be useful for MNIST is used? The paper currently has an experiment performed on MNIST (with no feature map or kernel), showing out-of-sample error in response to number of passes. However, the experiment does not validate any of the assumptions or explain whether the observed outcome was predicted by the paper's theory. Finally, Section 4.2 describes in its final paragraph a setting where multi-pass SGD performs as well as the empirical risk minimizer, and that both are minimax optimal. Do we know whether a single-pass cannot achieve this result as well? If so, then this paragraph seems to imply, independently of the multi-pass SGD result, a separation between single-pass SGD and empirical risk minimization, which would be a result worth highlighting or explaining on its own. If not, then seeing as how both single-pass and empirical risk minimization work, it seems less surprising that multi-pass (which is in a sense between the two) works as well. Either way, it seems important to clarify which of the two scenarios holds. ---- Edited after author response: It appears that, overall, the author response answers/addresses most of my questions/concerns above.

Reviewer 2



This paper presents generalization bounds for SGD with multiple passes over the dataset (with large constant stepsizes) for the least squares regression problem. The result is interesting and is consequential to practice. The paper goes into the details of explaining the tradeoffs of multi-pass learning based on the spectrum of eigenvalue decay and the hardness of the problem measured by the optimal predictor norm. I am unable to quite appreciate these tradeoffs in context of practice (in the sense of which regime(s) are the most practically useful or most common); so clarifications of this sort would certainly improve understanding for interested readers. The proof technique especially for tracking the averaged SGD iterate to the gradient descent iterate (in the appendix) was pretty neat and goes in a line similar to the technique of Bach and Moulines (2013). A comment regarding related work: Clearly the authors exhibit a very good understanding about the state of the art in providing generalization bounds for SGD (or its accelerated variants) for least squares (and beyond). Yet, it is surprising that they do not mention several recent papers that provide the strongest known bounds dedicated for least squares stochastic approximation (and beyond): [1] SGD and randomized Kaczmarz [Needell et al 2013], [2] SGD for least squares [Defossez and Bach, 2015] [3] Streaming SVRG [Frostig et al 2015], [4] SGD for least squares [Jain et al. 2016], [5] Accelerated SGD for least squares [Jain et al 2017]. The difference from what I see is that these papers employ strong convexity (note here n>d) - nevertheless, they pretty much define the current state of non-asymptotic generalization bounds for this problem and are a part of the context of this problem. It would help to compare and contrast this result against ones established in the above papers. ***post-rebuttal*** As for clarification regarding notions of problem hardness, yes, this certainly will help readers understand the significance of these results in practice. These papers does not deal with multi-pass SGD, but they define the current state of the art for SGD (and their accelerated variants) rates in certain problem hardness regimes.

Reviewer 3



Summary: The paper considers the convergence properties of multiple-passes averaged SGD under more natural (and optimal) choices of step sizes. The authors address a regime of parameters where performing more than #data iterations is provably necessary to obtain optimal statistical efficiency. Moreover, the upper bound provided in the paper is matched by known lower bounds (at least, for a significant fraction of the identified regime). The theoretical findings are then claimed to be supported by experiments conducted on synthetic data and MNIST. Evaluation: Bridging the gap between the ML practice of performing multiple-passes averaged SGD (which is beneficial in many practical applications) and our theoretical understanding is a fundamental and intriguing question. The paper makes an important contribution in addressing this gap by identifying a (non distribution-free) regime where various regularity assumptions on the data provably justify multiple-passes. The analysis mainly revolves around on an extension of a previously-established connection between full-gradient descent and the averaged SGD iterates (which could also potentially form a weakness of this technique). The predictions follow by the analysis are claimed to be confirmed by experiments conducted on synthetic data and MNIST, but I'm missing some details here which lowered my overall score (see below). Lastly, I find this paper relatively accessible and easy-to-follow. Comments: P1, L35 - can you elaborate more on 'using these tools are the only way to obtain non-vacuous' or provide a reference (or forward-ref for the minimax section if that was your attention)? P2, L52 - Can you provide a pointer for 'best prediction performance... as..'? P2, L53 - Do you have more intuition as to *what* makes the 'easy problems' easy (and vice versa) in terms of the interplay between alpha and r? P2, L65 - 'parametric models larger dimensions' seems gramtically incorrect. P3, L87 - The way the paragraph right after assumption 3 is pharsed seems to render this assumption irrelevant. More in this context, assumption 4+5 always hold for finite-dimensional feature spaces. In general, the split into the finite- and infinite-dimensional cases seems to be referred to by the paper incoherently. P6, L213 - 'lead' P6, L218 - 'let' P7, L257 - by 'misspecified', do you mean underspecified? P7, L270 - Not sure I completely understand how you estimate the optimal t_*(n) (this made it somewhat tricky for me to judge the soundness of the results). Did you use validation-set based tests? Also, what is the behavior of the excess risk when t's are taken to be larger than t_*(n)?